


# Source characterization of Highly Oxidized Multifunctional Compounds in a Boreal Forest Environment using Positive Matrix Factorization

Chao Yan[1], Wei Nie[2,1], Mikko Äijälä[1], Matti P. Rissanen[1], Manjula R. Canagaratna[3], Paola Massoli[3], Heikki Junninen[1], Tuija Jokinen[1*], Nina Sarnela[1], Silja Häme[1], Siegfried Schobesberger[1**], Francesco Canonaco[4], Andre S. H. Prevot[4], Tuukka. Petäjä[1], Markku Kulmala[1], Mikko Sipilä[1], Douglas R. Worsnop[1,3], and Mikael Ehn[1].

[1] Department of Physics, University of Helsinki, Helsinki, 00140, Finland
[2] Joint International Research Laboratory of Atmospheric and Earth System Sciences, Institute for Climate and Global Change Research & School of Atmospheric Sciences, Nanjing University, Nanjing, 210046, China
[3] Aerodyne Research, Inc., Billerica, MA 01821, USA
[4] Laboratory of Atmospheric Chemistry, Paul Scherrer Institute, Villigen, 5232, Switzerland
*now at Department of chemistry, University of California, Irvine, CA, 92617, USA
** now at Department of Atmospheric Sciences, University of Washington, Seattle, Washington 98195, USA.

*Correspondence to*: C. Yan (chao.yan@helsinki.fi) and W. Nie (niewei@nju.edu.cn)

**Abstract**

Highly oxidized multifunctional compounds (HOMs) have been demonstrated to be important for atmospheric secondary organic aerosols (SOA) and new particle formation (NPF), yet it remains unclear which the main atmospheric HOM formation pathways are. In this study, a nitrate ion based Chemical Ionization Atmospheric-Pressure-interface Time-of-flight mass spectrometer (CI-APi-TOF) was deployed to measure HOMs in the boreal forest in Hyytiälä, southern Finland. Positive matrix factorization (PMF) was applied to separate the detected HOM species into several factors, relating these "factors" to plausible formation pathways. PMF was performed with a revised error estimation derived from laboratory data, and this approach was validated by mathematical diagnostics of the PMF solutions. Three factors explained the majority (>95%) of the data variation, but the optimal solution found six factors, including two nighttime factors, three daytime factors, and a transport factor. One nighttime factor is almost identical to laboratory spectra generated from monoterpene ozonolysis, while the second likely represents monoterpene oxidation initiated by $NO_3$. The exact chemical processes forming the different daytime factors remain unclear, but they all have clearly distinct diurnal profiles, very likely related to monoterpene oxidation with a strong influence from NO, presumably through its effect on peroxy radical ($RO_2$) chemistry. Apart from these five "local" factors, the sixth factor is interpreted as a transport related factor. These findings improve our understanding of HOM production by confirming current knowledge and inspiring future research directions, and provide new perspectives on using factorization methods to understand short-lived atmospheric species.

## 1. Introduction

Large amounts of volatile organic compounds (VOCs) are emitted into the atmosphere from both biogenic and anthropogenic sources (Atkinson and Arey, 2003). These VOCs are oxidized in the atmosphere, which leads to



thousands of structurally distinct products, containing many functionalities (Hallquist et al., 2009). A subset of these
products become highly oxidized multifunctional compounds (HOMs, Ehn et al., 2012) and, although generally
considered a minor pathway in VOC oxidation, they play a crucial role in atmospheric aerosol formation (e.g. Kulmala
et al., 2013; Ehn et al., 2014; Jokinen et al., 2015), and thereby both air quality (Nel, 2005) and climate (IPCC 2013).

The existence of HOMs had been suggested by model studies, which assumed that a fraction of the VOC oxidation
products was effectively non-volatile (Spracklen et al., 2011; Riipinen et al., 2011).  Only recently, with the
development of the APi-TOF (Junninen et al., 2010) and later the CI-APi-TOF (Jokinen et al., 2012) has it been
possible to directly detect these HOMs (Ehn et al., 2012; Ehn et al., 2014), with subsequent studies dedicated to
understand the atmospheric implications of HOMs. Systematically investigation of new-particle formation (NPF)
events observed at the SMEAR II station in southern Finland, suggested a key role of HOMs in NPF (Kulmala et al.,
2013). Further laboratory studies have confirmed this finding. Schobesberger et al. (2013) showed that HOMs can
participate in the initial steps of NPF by stabilizing sulfuric acid, and the inclusion of this mechanism significantly
improves the model prediction of particle number concentration  (Riccobono et al., 2014). Ehn et al. (2014) have
simulated HOM formation with $O_3$ and α-pinene (the most abundant biogenic VOC in high latitudes), and shown that
these HOMs can explain the majority of the observed particle growth from 5 nm up to 50 nm at SMEAR II. Though
the molar yield of HOMs is only a few percent depending on the VOC structure and oxidant, a global model suggested
HOMs play a crucial role in secondary organic aerosol (SOA) burden and CCN concentrations (Jokinen et al., 2015).

As HOMs are important compounds linking VOCs to SOA, quantitative simulation of SOA formation requires
detailed understanding of HOM formation. According to current knowledge, the formation of HOMs consists of two
consecutive processes: 1) VOC oxidation forming peroxy radicals ($RO_2$) able to auto-oxidize through intramolecular
H-abstraction, leading to multiple $O_2$ additions; and 2) termination reactions, which terminate the auto-oxidation by
converting $RO_2$ radicals into closed-shell molecules. Ehn et al. (2014) successfully simulated ambient nighttime HOM
spectra by adding $O_3$ and α-pinene into a chamber, indicating the importance of that $O_3$-initiated oxidation and the
following multi-step H-shift reactions (auto-oxidation). Jokinen et al. (2014, 2015) later expanded the HOM
observations to a broader group of VOC precursors and oxidants ($O_3$ and OH). Similar processes has been confirmed
for the $NO_3$-initiated monoterpene oxidation investigated by Boyd et al. (2015) with chemical ionization using I⁻ as
the reagent ion. Termination reactions occur in competition with further auto-oxidation, and may even prevent it
altogether. In the atmosphere, $RO_2$ termination may happen by reacting with partners ("terminators", i.e. hydroperoxyl
radical ($HO_2$), $RO_2$, $NO_x$), or undergoing self-termination (Orlando and Tyndall, 2012). The large variety of
terminators leads to critical branching steps in the atmospheric oxidative pathways, eventually resulting in a large
number of different HOM molecules. Despite the new insights acquired from recent chamber studies, HOM formation
in the complex atmosphere remains poorly understood. One of the fundamental reasons is the lack of robust methods
to analyze the complicated ambient data (e.g. mass spectra containing >>100 molecular ions) and to link ambient
observations and chamber studies.



Positive matrix factorization (PMF) (Paatero and Tapper, 1994) allows for time resolved mass spectra to be expressed
as a linear combination of a finite number of factors, assuming that the factor profiles are constant and unique (Ulbrich
et al., 2009). Since this method does not require a-priori information about the factors, it is an ideal technique for
extracting information from ambient measurements where the detailed chemistry, sources, and atmospheric processes
are complex. PMF analysis of aerosol mass spectra, for example, has been widely utilized to identify multiple primary
organic aerosol sources (i.e. vehicle emissions, biomass burning, cooking) and to characterize secondary organic
aerosol aging via factors with varying volatilities and oxidation levels (Lanz et al., 2007; Ulbrich et al., 2009; Ng et
al., 2010; Jimenez et al., 2009; Zhang et al., 2011). PMF has also been applied to analyze time-resolved ambient proton
transfer reaction mass spectrometer (PTR-MS) measurements of organic species in the gas phase (Vlasenko et al.,
2009; Yuan et al., 2012) and to analyze combined AMS-PTR-MS datasets (Slowik et al., 2010; Crippa et al., 2013).

In this work, we report the first success of utilizing PMF on CI-APi-TOF data. We examine the degree to which the
PMF factors represent the dominant HOM formation pathways at the observation site, and attempt to validate the
retrieved factors by comparison to existing chamber data and correlation with other co-located measurements. Our
results link the ambient measurement to previous chamber studies, and identify needs for future research efforts in
this area. This work also provides new perspectives on using PMF to understand the variation of short-lived species,
e.g. HOMs.

**2.    Measurement**
**2.1 Site description**
In this study, the measurement data was obtained at boreal forest research station SMEAR II located in Hyytiälä,
southern Finland (Hari and Kulmala, 2005). The station is surrounded by boreal conifer forest and is described as a
rural continental background measurement site (e.g. in (Manninen et al., 2010)). The nearest large cities are Tampere
(around 60 km to South-West, 213 000 inhabitants) and Jyväskylä (around 100 km to North-East, 131 000 inhabitants).
SMEAR II is a rural site, but sometimes polluted air masses reach the site causing relatively high aerosol loadings and
high concentrations of gas-phase pollutants. Typical pollutants are from forest fires in Russia, biomass burning from
eastern Europe, Tampere urban plume or a nearby sawmill (southeast of SMEAR II) (e.g. Liao et al., 2011; Ulevicius
et al., 2015). Ambient meteorological conditions such as temperature, relative humidity (RH), solar radiation, wind
speed and direction, particle concentration and size distribution, as well as concentrations of aerosol particles and
several trace gases, e.g. carbon dioxide ($CO_2$), carbon monoxide (CO), sulfur dioxide ($SO_2$), nitrogen oxides ($NO_x$)
and ozone ($O_3$), are continuously monitored at the station.

**2.2 Measurement of oxidized organic compounds**
A nitrate ion ($NO_3^-$) based Chemical Ionization Atmospheric-Pressure-interface Time-of-flight mass spectrometry
(CI-APi-TOF) was deployed to measure the highly oxidized organic compounds as well as sulfuric acid in an intensive
observation period in April-May, 2012. This state-of-the-art instrument can sensitively and selectively measure many
HOMs with high oxygen to carbon ratio. Instrument and measurement details have been described elsewhere





(Junninen et al., 2010; Jokinen et al., 2012). The mass spectra were analyzed with the tofTools package developed by
Junninen et al. (2010). The quantification of any compound X is calculated as
$[X] = \frac{\sum_{i=0}^{2}[(HNO_3)_i(NO_3^-)(X) + (HNO_3)_i(X-H)^-]}{\sum_{i=0}^{2}(HNO_3)_i(NO_3^-)} \times C_X$   (Eq. 1)
Here [X] is the concentration of the neutral compound to be quantified, the numerator on the right hand side is the
sum of all detected ions containing the compound X (either by deprotonation or as an adduct with $NO_3^-$),
the denominator is the sum of all reagent ion signals, and $C_x$ is the calibration coefficient representing the detection
sensitivity for compound X. For the measurement of total HOMs, we summed up all signals within the mass range of
201~650 Th excluding some known instrumental background peaks. As suggested by Ehn et al. (2014), the calibration
coefficient for HOMs is assumed equal to the value used for sulfuric acid within 50% uncertainty. The calibration
coefficient reported by Jokinen et al. (2012) is used in this work, as the tuning of the instrument and the geometry of
the sampling tube were similar.

**2.3   Positive matrix factorization(PMF)**
**2.3.1      Working principle and advantages of PMF**
PMF is a well-established algorithm based on the work by Paatero and Tapper (1994). This receptor model is useful
for solving functional mixing models when the source number and source profiles are unknown. It fundamentally
works on an assumption of mass conservation so that a mass balance analysis can be used to identify and apportion
sources of the detected species in the atmosphere. The most important feature that distinguishes PMF from other
receptor modeling (e.g. principal component analysis) is that it applies a least-squares algorithm that accounts for data
uncertainties. It also constrains the solutions to the non-negative subspace so that they are environmentally reasonable.
Due to these advantages, this algorithm is widely-used for source apportionment, especially on aerosol mass spectra
(Zhang et al., 2011). The PMF analysis in this work uses the IGOR based analyzing interface SoFi (solution finder,
version 5.2) as described in Canonaco et al. (2013).

In PMF, the mass balance can be described as
$Y = GF + E$ (Eq. 2)
Matrix Y is an m×n matrix, usually representing m measurements (in time or samples) of n variables. The sizes of the
factor matrices G and F are m×p and p×n, respectively, where p is the number of factors. In practice, the matrix G is
the time series of the p factors representing the source strength, and matrix F is the profiles of the p factors showing
the variable distributions of the sources. Matrix E is the residual left unexplained by the p factors. It should be noted
that the value of p is not pre-fixed, and determination of the value will be based on the interpretability of the solutions.

The PMF algorithm seeks to minimize Q. Q is the sum of squared residual weighted by the inverse of their respective
measurement uncertainty, which can be described as
$Q = \sum_{i=1}^{m}\sum_{j=1}^{n}(\frac{E_{ij}}{S_{ij}})^2$   (Eq. 3)





Here $S_{ij}$ is the error representing the estimated measurement uncertainty of element j at time i, and $E_{ij}$ is the
corresponding residual. In this work, the uncertainty was estimated from laboratory data, which will be discussed in
section 2.3.3. Data points where $E_{ij} \gg S_{ij}$ have a large influence on the model iteration, and this needs to be reduced
or removed by the model. A robust mode is applied to eliminate the strong outliers determined by α, meaning that any
data points yielding $E_{ij}/S_{ij} > α$ will be reduced to this threshold:
if $\left|\frac{E_{ij}}{S_{ij}}\right| > α, \left|\frac{E_{ij}}{S_{ij}}\right| = α$ ; (Eq. 4)
where the value of α is a free parameter can be determined by the user, and a value of 4 was suggested by Paatero et
al. (1997).

Ideally, the modeled Q value should eventually approach to the expected Q values ($Q_{exp}$), which is equal to the degree
of freedom of the model solution (n×m – p(n+m)). For mass spectra data (e.g. AMS spectra, CI-APi-TOF spectra), it
is roughly equal to the size of the matrix:
$Q_{exp} = n×m – p(n+m) \approx (n×m)$ (Eq. 5)

**2.3.2    Data matrix**
The nitrate ion based CI-APi-TOF selectively measures HOMs with a ~4000 Th/Th resolving power. In principle, this
resolution allows us to fit peaks and in some cases resolve peaks with different composition at the same unit mass.
However, the quality of the peak fitting strongly depends on mass calibration of the spectrum and the smoothness of
the peaks. We found that the mass calibration may shift by 5 ppm by using data with 5-minute integration time, and
some HOM peaks are not smooth enough due to the weak signals. Fitting the peaks beforehand in such circumstances
may introduce extra and non-uniform uncertainties that are difficult to estimate. Therefore, the data matrix used in
this work is in unit-mass resolution, and peak fitting was performed afterwards to identify the elemental formula of
peaks. Some examples of peak fitting are provided in Fig. S9. The mass range of 201 – 650 Th was selected for PMF
analysis, which covers most of the detectable HOMs. We continuously collected data from Apr. 4[th] to May 7[th], 2012,
with very few missing time points due to instrumental issues. These raw data were averaged into 5-min time resolution,
and a total number of 9084 mass spectra were then obtained. Thus, the final data matrix is in the size of 9084 (samples)
×450 (variables).

**2.3.3    Error matrix estimation**
Due to the abovementioned algorithm principle, the estimation of error matrix ($S_{ij}$) is crucial. Suggested by Polissar
et al., (1998), the error matrix in this work was estimated as Eq. 6 shown below:
$S_{ij} = σ_{ij} + \frac{DL}{3}$ (Eq. 6)
Here, $σ_{ij}$ is the analytical uncertainty of a certain data point, and the DL is the limit of detection of variables. We apply
a constant DL for all variables detected by the instrument, determined as the standard deviation of time variation in
'ion-free' mass ranges (see supplementary information Fig. S3). The $σ_{ij}$ was estimated based on the assumption that
the counting statistics follow the Poisson distribution (Allan et al., 2003):


$\sigma_{ij} = a \frac{\sqrt{I}}{\sqrt{t_s}}$   (Eq. 7)
I is the signal strength (ions/second) of the ion, $t_s$ is the integration time in seconds, and $a$ is a factor accounting for
the fact that a single ion will generate a Gaussian-shaped pulse in the detector, rather than a single peak. Error
estimation for aerosol mass spectrometer (AMS) data usually applies 1.2 for $a$ value (Allan et al., 2003). We
determined the proper $a$ value for CI-APi-TOF data with a set of laboratory experiments. The schematic of the
corresponding experimental setting is provided in the supplementary information (Fig. S1). A temperature controlled
permeation tube was set in front of the chemical ionization inlet (CI-inlet). A 100 mlpm (milliliter per minute) $N_2$ gas
served as carrier gas flowing through the permeation tube, which then was diluted with 10 lpm (liter per minute) $N_2$
before entering the CI-inlet. The experiments were run under the following conditions:
1)   two identical permeation tubes were used, filled with perfluoro-butanoic acid ($CF_3(CF_2)_2COOH$) and

193         perfluoro-nonanoic acid ($CF_3(CF_2)_7COOH$), respectively;

2)   for each chemical, temperature was changed every hour to creat multiple steps of stable signals (Fig. S4);
3)   A certain chemical and temperature was repeated twice for different instrumental tunings. This is to check if

196         $a$ and DL are influenced by instrument tunings.

With the stable signals during these experiments, the error was fitted to the signal strength based on Eq. 6. Detailed
information and discussion are provided in supplementary information Section S1. Briefly, the results show that the
DL is about 0.105 ion/s, stable and independent of temperature and instrument tuning; the $a$ value was estimated to
be 1.3, which is very close to the value (1.2) suggested for AMS data. For 5-min data, the equation of error estimation
is shown below:
$S_{ij} = 0.074 \sqrt{|Y_{ij}|} + 0.035$   (Eq. 8)

We also proposed a different statistical method based on ambient data (see Supplementary information section S2). A
comparison of different uncertainty estimation schemes is shown in Fig. 1, where the red curve denotes the revised
error estimate in this work, the blue one is the customary estimate for AMS data, and the gray area represents the error
from ambient data by using a different estimation scheme. Within the fitting uncertainty, all three estimates agree well.

As suggested by Paatero et al (2003), two more steps were deployed to further modify the error estimation. 1) for
variables that are below the DL, we fixed the concentration as 1/3 DL and the corresponding uncertainty as DL, which
will cause a smaller weight for these data points in the algorithm. 2) a down-weight scheme was also applied for
variables with a low signal-to-noise ratio (SNR), i.e. $Y_{ij}/S_{ij}$, which further increases the error by 2 and 10 folds for
"weak" (SNR<2) and "bad" (SNR<0.2) signals, respectively.
**3   Data overview**

The data were collected at the SMEAR II station from April 4 to May 7, 2012. Fig. 2 shows the time series of
meteorological conditions (i.e. global radiation, UVA, UVB, and temperature), concentration of trace gases (NO, $NO_x$,
$O_3$, $SO_2$), sulfuric acid (SA) concentration, and total HOM concentration. Looking at global radiation or UVA and





UVB intensity (global radiation > 400 W m$^{-2}$ or UVA > 15 W m$^{-2}$, UVB > 0.2 W m$^{-2}$), 78 % (26 out of 33) of the
days in the measurement period had strong photochemical activity, the rest being cloudy days when photochemistry
was significantly suppressed.  From Apr. 9$^{th}$ to Apr. 12$^{th}$, air mass analysis using backward Lagrangian particle
dispersion model (LPDM) (Ding et al., 2013) indicates that, the measurement site was influenced by a polluted plume
originating from eastern Europe (Fig. S10); clear elevations of anthropogenic pollutants, such as $SO_2$ and $NO_x$ were
observed. During the entire period, the measured sum of HOM concentration exhibited clear diurnal variations, with
notably higher levels in the daytime. Note this contrasts with lower daytime monoterpene concentrations trend that
are typically observed VOCs at the site (Rantala et al., 2014), consistent with photochemical HOM production during
daytime.

Apart from the variable concentrations, spectral differences between daytime and nighttime are also evident. The
averaged spectra are presented in Fig. 3a, where night- and daytime spectra are shown below and above the zero line,
respectively. As monoterpenes ($C_{10}H_{16}$) are known as the dominant precursors for HOMs at this location (Ehn et al.,
2012), we divided the mass range (201 – 650 Th) into three sub-ranges: 1) 201 – 290 Th for lighter HOM compounds,
mostly containing 3 to 7 carbons; 2) 290 – 450 Th for HOM 'monomer' products, mostly fitting the general formula
$C_{9-10}H_{14-16}O_{7-13}N_{0-2}$; and 3) 450 – 650 Th for HOM 'dimer' products with the general formula $C_{16-20}H_{28-32}O_{9-19}N_{0-2}$.
Expanded mass spectra are shown in Fig. 3b, c, and d, where some major peaks are labeled with their elemental
formula. The lighter HOMs show notably elevated concentrations in the daytime. HOM monomers in the nighttime
spectrum are similar to those reported in previous chamber studies (e.g. Ehn et al., 2014), whereas major peaks in the
daytime are very likely organo-nitrates. These plausible organo-nitrates were identified with high yields when mixing
monoterpenes, $O_3$, and $NO_x$ in the chamber  (Ehn et al., 2014; Jokinen et al., 2014), and they are also suggested to be
important to NPF (Kulmala et al., 2013). Higher signals of HOM dimers are observed in the nighttime, with many
major peaks similar to those have been reported by Ehn et al. (2014). However, there are also peaks likely containing
nitrogen, which are produced through different reaction pathways.

Below, all elemental formulas for molecules containing N atoms will be expressed as $NO_3$ groups, since such organo-
nitrate functionality is the only expected form of $NO_3$ (-ONO2) in HOM species.

**4 Results and discussion**
**4.1 Evolution of PMF solutions**
Since the PMF analysis is performed without any a priori knowledge, the choice of the proper number of factors is the
most critical decision towards interpreting the PMF results. Choosing the best factor number is a compromise. More
factors give the model more freedom to explain subtle variations of the data but, on the other hand, too many factors
can force the model to split a physically meaningful factor into unrealistic ones. In this work, PMF analysis was
initially done for two factors, and followed with a step-wise addition of one factor until the additional factor could no
longer be interpreted based on their unique mass spectral features or comparisons of their time trends with auxiliary
data. Fig. 4 shows the average contribution of PMF solutions to HOM concentration assuming two to seven factors.



Our main analysis focuses on the 6-factor solution, but a short discussion of factor evolution is included below (factor
profile and time series is shown in Fig. S11).
The two factor solution leads to distinct day- and nighttime factors. The spectral difference is also obvious: daytime
factor contains more light HOM molecules but few HOM dimer products, while the nighttime factor contains very
few light HOM molecules but most of the HOM dimer products. In addition, peaks with odd masses, which are likely
nitrate containing HOMs, dominate the daytime factor, while the major peaks in the nighttime factor have even masses
and are unlikely to contain organic nitrogen.
In the 3-factor case, the profile of two factors (daytime factor and nighttime factor) are more or less the same as those
in the 2-factor case, while the new factor is featured by a prominent peak at 201 Th, which is identified as nitrophenol
($C_6H_5NO_3$), although this species is detected as an adduct with $NO_3^-$. Since the new factor exhibits a weak diurnal
cycle, we temporally name it with its prominent peak, "201 Th factor".
In the 4-factor solution, the daytime factor in the 3-factor case splits into two new factors, termed daytime type-1 and
daytime type-2, respectively. Their diurnal patterns are different – the daytime type-1 factor starts to increase at 4am
and reaches the peak at 10am, while the daytime type-2 factor starts to increase at around 6am, and reaches the peak
around 11am – 3pm. The major peaks in both new factors are organo-nitrates but in different masses – 355 Th
($C_{10}H_{15}O_6NO_3$) and 387 Th ($C_{10}H_{15}O_8NO_3$) are the most prominent peaks in the daytime type-1 factor, and 339 Th
($C_{10}H_{15}O_5NO_3$) is the highest peak in the daytime type-2 factor.
Introducing a fifth factor retrieves a third daytime factor. The other two daytime factors remain similar to those in the
4-factor solution in respect to their diurnal patterns and major peaks, with their contributions to total HOM
concentration reduced from 15 % and 23 % to 11 % and 20 %, respectively (Fig. 4). The contribution of the "201 Th
factor" also has a pronounced decrease from 34 % to 24 % (Fig. 4), and its diurnal pattern has a clear change - peaking
time changed from 12 am to 9 am. The new daytime type-3 factor starts to increase at 6 am in the morning and reach
its peak value at 2 pm. Fingerprint peaks in this factor are 213 Th ($C_3H_5O_3NO_3$), 241 Th ($C_4H_5O_4NO_3$), 255 Th
($C_5H_7O_4NO_3$), 269 Th ($C_6H_9O_4NO_3$), and 281 Th ($C_7H_9O_4NO_3$).
The six factor solution separates nighttime factor into two different factors, namely nighttime type-1 and nighttime
type-2, with the remaining factors are almost unchanged with respect to the 5-factor solution (Fig. 5 and Fig. 6). Both
new factors show elevated concentrations in the nighttime. The dominant peaks in the nighttime type-1 factor contain
even masses in both HOM monomer and dimer mass ranges. In the nighttime type-2 factor, on the other hand, more
intense odd-mass peaks are present, such as 403 Th ($C_{10}H_{15}O_8NO_3$) and 419 Th ($C_{10}H_{15}O_9NO_3$) in the monomer range,
as well 523 Th ($C_{20}H_{31}O_8NO_3$), 554 Th ($C_{20}H_{32}O_6(NO_3)_2$), and 555 Th ($C_{20}H_{31}O_9NO_3$) in the dimer range.
When seven factors are assumed, an additional daytime type factor appears. The new factor contains peaks that are
mostly identified as nitrogen-containing organic compounds with 4-10 carbon atoms. Since there is no strong



correlation with any independent tracer, we choose to limit our further analysis to the six-factor solution. Note, without
such correlations, it is not possible to distinguish the identification of "real" factors.

**4.2 Mathematical diagnostics of PMF solutions**
Mathematical diagnostics are a key criterion in evaluating PMF model performance. The mathematical diagnostics in
this work include the $Q/Q_{exp}$ value, the distribution of Q over time and variables, the fraction of explained variation in
the data, and consistency of seed runs.

The $Q/Q_{exp}$ provides the most direct reflection of the goodness of error estimate and the validity of PMF results, as the
model runs to seek for the minimal $Q/Q_{exp}$ value. A too large (e.g. >10) or too small (e.g. <0.1) $Q/Q_{exp}$ may suggest a
bias of the uncertainty estimation. Fig. 7 shows the change of $Q/Q_{exp}$ and the explained variation as a function of factor
number. From two to seven factors, $Q/Q_{exp}$ decreases stepwise from 2.44 to 0.76. The closeness to unity indicates that
the estimated error is appropriate for the model. As suggested by Ulbrich et al. (2009), the decreasing trend of $Q/Q_{exp}$
is useful to determine the minimum factor number, as a large decrease in $Q/Q_{exp}$ indicates the additional factor may
explain a large fraction of unaccounted variability in the data. As shown in Fig. 7, the third factor significantly
decreases the $Q/Q_{exp}$ value. As mentioned in Section 2.3.1, the robust-mode PMF guaranteed that some strong outliers
would not distort the model algorithm. However, the distributions of Q over time and variables are examined in order
to help identify variables and time steps that were not fit well (see supplementary information Section S3).

The explained fraction of data variation with regard to factor number is also shown in Fig. 7. With two factors, the
model explains about 92 % of the data variation, and adding the third factor largely increases the explained fraction
to 95 %. The explained fraction also rises to 97 % when adding the sixth factor, suggesting the separation of the two
nighttime type factors is significant. A slight reduction of explained variation is observed when seven factors are
assumed, suggesting the 7-factor PMF is not an appropriate PMF solution.

In order to evaluate the consistency of the PMF results, we run the PMF algorithm from five different random starting
points for each number of factors (seed runs, (Paatero, 2007)). As shown in Fig.7, the five seed runs for each factor
number show good consistencies in both $Q/Q_{exp}$ and explained variation, indicating the small model uncertainty. The
only exception is the 5-factor PMF, where the results in five seed runs show two groups with small discrepancies.
This can indicate that there are likely two factorizations that generate equally valid solutions, suggesting that one more
factor is required to resolve both factorizations.

**4.3 Interpretation of PMF results**
The mathematical diagnostics characterize the technical aspects of PMF. However, they are not guaranteed to give
the most realistic solution. PMF is a descriptive model, thus the "interpretability" or "meaningfulness" is the most
critical criterion in determining the best solution. Interpretation of PMF results needs careful examination of each
retrieved factor, which usually requires many considerations, including:





1. Comparison between the profile of retrieved factors and reference spectra from laboratory studies. The uncentered correlations (UC, Eq.9, Ulbrich et al. 2009) is used to quantitatively assess the similarity:

$$UC = \frac{x \cdot y}{\|x\| \, \|y\|} \quad \text{(Eq. 9)}$$

where x and y denote a pair of time series or factor profile as vectors. In fact, as a new measurement technique, only a few of reference spectra have been reported for monoterpene oxidation (Jokinen et al., 2014; Ehn et al., 2014; Mutzel et al., 2015);

2. Identification of key molecules as specific fingerprints of factors, as listed in Table 1. These molecules are chosen either if they are the most visible ones in the profile or if they are mostly (usually >70%) allocated to one specific factor. This method is rationalized by the fact that much molecular information is retained in the spectra, which helps to deduce the plausible reaction pathways;

3. Temporal correlation of factors with other tracers which represent specific sources or atmospheric processes;

4. Other information such as meteorology (e.g. air mass trajectories).

Based on these considerations, we concluded that the PMF solution with six factors is the optimal solutions. Fig. 5 shows the spectra of the six factors, and their diurnal patterns are shown in Fig. 6, together with some relevant trace gases and meteorological parameters. It should be noted that all the mass spectra and diurnal profiles are very distinct, indicative of a realistic PMF solution. In the following sub-sections, each factor is discussed in detail.

**4.3.1 Nighttime factors**

*Nighttime type-1 factor*

The nighttime type-1 factor is the largest contributor to nighttime HOM concentration. It exhibits elevated intensity during 8pm – 4am and is less intense (about five times lower) in the daytime. The major peaks in this factor are identified as $C_{10}H_{14-16}O_{6-13}$ and $C_{19-20}H_{28-32}O_{10-18}$. As shown in Fig. 8, the profile of this factor is very similar to the reference spectrum from previous laboratory studies reported by Ehn et al. (2014), where only ozone and α-pinene were mixed. It should be noted that, in the atmosphere, there is always a mixture of monoterpenes likely contributing to these signals, in contrast to a single monoterpene precursor was used in the chamber experiments. Also humidity and temperature were typically different and changing constantly, and all these together can explain the minor difference in individual peak intensities, for example, 372 Th ($C_{10}H_{14}O_{11}$) and 389 Th ($C_{10}H_{15}O_{12}$) are higher in the reference spectrum than in the factor profile. The coefficient of uncentered correlations between the factor profile and the reference spectrum was calculated to be 0.91, confirming the high similarity between them. Thus, the source of this factor is very likely the ozonolysis of monoterpenes.

*Nighttime type-2 factor*

The diurnal variation of the nighttime type-2 factor has a similar pattern to that of the nighttime type-1 factor. Its intensity is about 30% percent of nighttime type-1 factor during the nighttime, and almost decreases to zero during the day (Fig. 6). To our knowledge, no reference spectrum that matches the profile of this factor (shown in Fig. 5) has been reported. However, a set of masses can represent a new fingerprint. Figure 9a shows these fingerprint peaks in the dimer range, which are categorized and marked in different colors. In general, the vast majority of compounds contain nitrogen and we divide dimer peaks in this factor into 6 groups according to their elemental composition, i.e.





$C_{20}H_{31}O_{7-15}NO_3$, $C_{20}H_{32}O_{4-11}(NO_3)_2$, $C_{19}H_{29}O_{6-13}NO_3$, $C_{19}H_{31}O_{8-11}NO_3$, $C_{18}H_{29}O_{8-11}NO_3$, and other non-nitrogen-
containing dimers. As dimers are closed-shell molecules, assumed to be formed through the reaction between two
peroxy radicals ($RO_2$) (Rissanen et al., 2014), the nitrogen atom(s) in the dimer molecule must come from its parent
$RO_2$ radical, suggesting $NO_3$-initiated oxidation. Note that the possibility of $NO_x$ involvement can be ruled out,
because when $NO_x$ reacts with $RO_2$, it either ends up with an organo-nitrate HOM monomer, or forms an alkoxy
radical (RO) so that the nitrogen atom will not retain in the molecule. The fractions of different groups are shown in
Figure 9b. About 61% of HOM dimers in this factor contain one nitrogen atom, suggesting that the major dimer
formation process involves reaction between two $RO_2$ radicals initiated by $NO_3$ and $O_3$, respectively. Also, about 22%
of these dimers contain two nitrogen atoms, meaning that both reacting $RO_2$ radicals are $NO_3$-initiated. The schematic
illustrations given below show two examples of dimer formation containing one nitrogen atom ($C_{20}H_{31}NO_{13}$, 555 Th
including $NO_3^-$) and two nitrogen atoms ($C_{10}H_{32}N_2O_{12}$, 554 Th including $NO_3^-$), respectively.
$O_3 + C_{10}H_{16} \xrightarrow{-OH\cdot} C_{10}H_{15}O_2^\cdot \xrightarrow{H-shift+O_2} \cdots \xrightarrow{H-shift+O_2} C_{10}H_{15}O_8^\cdot \xrightarrow{H-shift+O_2} C_{10}H_{15}O_{10}^\cdot$   (1)
$NO_3 + C_{10}H_{16} \rightarrow C_{10}H_{16}NO_3^\cdot \xrightarrow{H-shift+O_2} \cdots \xrightarrow{H-shift+O_2} C_{10}H_{16}O_4NO_3^\cdot$   (2)
$C_{10}H_{15}O_8^\cdot + C_{10}H_{16}O_4NO_3^\cdot \rightarrow C_{20}H_{31}O_{10}NO_3 + O_2$   (3)
$C_{10}H_{16}O_4NO_3^\cdot + C_{10}H_{16}O_4NO_3^\cdot \rightarrow C_{20}H_{32}O_6(NO_3)_2 + O_2$   (4)
As $NO_3$ is involved in the formation of more than 80% dimer molecules, the nighttime type-2 factor is likely
representing monoterpene oxidation by $NO_3$.

***Comparison of the two nighttime factors***
As mentioned above, the nighttime factors are interpreted as representing nighttime oxidation of monoterpene initiated
by the two major nocturnal atmospheric oxidants – $O_3$ and $NO_3$, respectively. Their nighttime patterns are similar,
exhibiting an increase at 8 pm and a decrease at 4 am in the next morning (Fig. 6). However, as the $O_3$ concentration
is relatively stable throughout day while $NO_3$ is much lower in the daytime, the $O_3$-initiated factor has finite level
during the daytime while the $NO_3$-initiated factor goes almost to zero. In general, the $O_3$-initiated factor is a larger
contributor than the $NO_3$-initiated factor, suggesting that $O_3$ is a more important nighttime oxidant for HOM formation
at this measurement location. However, as shown in Fig. 10a, during the period (from Apr. 9[th] to Apr. 12[th]) when
polluted air masses containing high $NO_x$ were transported to this area, the $NO_3$-initiated oxidation was significantly
enhanced and became dominating. Since the $NO_3$ chemistry could be one important pathway of forming HOMs, future
laboratory study of this reaction channel is required.

**4.3.2 Daytime factors**
The interpretation of daytime factors is more difficult, likely reflecting more complex daytime photochemistry.
Nevertheless, certain conclusions can be drawn from spectral characteristics and temporal behavior of the three
daytime factors.
***Daytime type-1 factor***
As shown in Fig. 6, this factor concentration starts to increase in the early morning (around 4 am), concurrent with the
increase of NO and the decrease of the two nighttime factors. The very similar temporal behavior of this factor and



NO (Fig. 10b) indicates that NO reaction is likely plays an important role in this factor. The two highest peaks in this
new factor are 355 Th ($C_{10}H_{15}O_6NO_3$) and 387 Th ($C_{10}H_{15}O_8NO_3$), which are likely formed through the reaction
between the two most abundant ($O_3$-initated) $RO_2$ radicals and NO, as shown below:
$C_{10}H_{15}O_8^{\cdot} + NO \rightarrow C_{10}H_{15}O_6NO_3$ $\qquad\qquad\qquad\qquad\qquad\qquad\qquad\qquad$ (5)
$C_{10}H_{15}O_{10}^{\cdot} + NO \rightarrow C_{10}H_{15}O_8NO_3$ $\qquad\qquad\qquad\qquad\qquad\qquad\qquad\qquad$ (6)
We hereby interpret this factor as products from $RO_2$ + NO reaction, which is also consistent with the observation that
no dimer HOMs are present because NO is the dominating $RO_2$ terminator in this pathway.

***Daytime type-2 factor***
The daytime type-2 factor is one of the main daytime HOM contributors. The major peak in this factor is found at 339
Th ($C_{10}H_{15}O_5NO_3$), the single highest organo-nitrate molecule observed at this site and the representative of daytime
HOMs previous reported by Kulmala et al (2013). Another major peak in this factor is 224 Th ($C_5H_6O_6$), possibly a
fragment of monoterpene oxidation as observed in laboratory experiments (e.g. Tröstl et al., 2016). Besides these
major peaks, this factor contains many other HOM monomer peaks.

This factor rises at around 5 am, reaches a maximum between 11 am and 3 pm (Fig. 6). Fig. 10c shows that the time
series of this factor and sulfuric acid are very similar. In cloudy days (UVB < 0.2 W m$^{-2}$), the intensity of this factor
is near zero. Note that this factor tracks sulfuric acid better than solar radiation. For example, the solar radiation was
similar on Apr. 7$^{th}$ and Apr. 8$^{th}$, whereas the factor's intensity was much lower on Apr. 8$^{th}$, similar to the variation of
sulfuric acid. Due to this reason, we interpret this factor as daytime oxidation of monoterpene controlled by OH,
though NO must also be involved because the single highest peak is an organo-nitrate. Also, note that the participation
of $O_3$ cannot be entirely excluded.

***Daytime type-3 factor***
The daytime type-3 factor shows maximum intensity in the afternoon around 2pm (Fig. 6). Fingerprint peaks in this
factor are organo-nitrate HOMs with smaller molecule weight, such as 213 Th ($C_3H_5O_3NO_3$), 241 Th ($C_4H_5O_4NO_3$),
255 Th ($C_5H_7O_4NO_3$), 269 Th ($C_6H_9O_4NO_3$), and 281 Th ($C_7H_9O_4NO_3$). Indicated by the smaller carbon number in
the molecules, these light HOMs may come from anthropogenic VOCs (i.e. benzene and toluene). However, this
possibility seems unlikely since the intensity of this factor does not show any significant increase during the period
with transported pollution (Apr.9 – Apr.12) when presumably benzene and toluene concentration were elevated.
Another possibility is that these compounds are fragments from the oxidation of larger VOCs (e.g. monoterpene), and
the presence of some HOM monomer peaks in this factor seems to support this assumption. This factor shows a good
correlation with UVB (see Fig. 10d, and Table 2), indicating the HOM formation pathway represented by this factor
is probably OH-initiated. Though the fingerprint peaks in this factor are organo-nitrates, the temporal variation of this
factor shows no dependence on NO concentration. Instead, it exhibits a similar pattern with temperature, as shown in
Fig. 10d. One possible explanation is that these small HOM molecules are relatively more volatile, so that their



aerosol-gas partitioning is strongly affected by temperature – higher temperature leads to less condensation and high
gas-phase concentration.

**4.3.3 Transport factor**
According to the mathematical diagnostics discussed in section 4.2, the third factor is important for the model to
account for a significant fraction of the variability in the ambient data. The only prominent peak in this factor is
nitrophenol ($C_6H_5NO_3$, 201 Th), a tracer for biomass burning suggested by previous studies (e.g. (Mohr et al., 2013)).
The temporal behavior of this factor is similar to $SO_2$, both showing a significant enhancement during period of Apr.9[th]
– Apr.12[th], when the measurement site was influenced by polluted air masses coming from eastern Europe (see Fig.
S10). We therefore suggest that this factor is a signature of transported pollution from biomass burning from
continental areas.

**4.4 Implication for atmospheric chemistry**
Theoretically, in the atmosphere, the formation pathway of HOM molecules involves addition of multiple $O_2$
molecules via autoxidation, including one oxidation initiation (by $O_3$, $NO_3$, or OH) and one termination reaction
(mainly by NO, $HO_2$, or $RO_2$). Each pathway serves as a HOM source, leading to distinct profiles of HOM products
for a specific VOC, with the overall HOM profile being a superposition of multiple pathways, depending on each
source intensity. In practice, the relative importance of these pathways is highly dependent on atmospheric conditions.
Table 2 lists suggested formation pathways for each factor, together with their correlation coefficients with other
relevant measurements. Nighttime type-1 and nighttime type-2 likely represent monoterpene oxidation initiated by
two major nighttime atmospheric oxidants, $O_3$ and $NO_3$, respectively. Indicated by high dimer concentrations from
$RO_2 + RO_2$ reaction, $RO_2$ is the main terminator for both of them, probably because $HO_2$ and NO concentration is
comparatively low in the nighttime. The daytime type-1 factor probably represents $O_3$-initiated oxidation followed by
NO termination. Though the exact chemistry producing the daytime type-2 factor is unclear, its clear dependence on
OH indicates the oxidative pathways have been shifted from dark chemistry ($O_3$ or $NO_3^-$ initiated oxidation) to
photochemistry (OH initiated oxidation). The initiator-terminator combinations that are not found in PMF solutions
may only have minor contributions to HOM production. For example, the combination of "OH-initiation" and "$RO_2$-
termination" may not exist, because in the daytime, NO and $HO_2$ are much more efficient in terminating $RO_2$.
Similarly, a pathway of "$NO_3$-initiation" followed by "NO termination" might be unlikely, probably because NO is
titrated by $O_3$ in the night, and $NO_3$ hardly exists in the daytime due to the photolysis.

**5. Conclusion**
HOMs have been confirmed by recent studies as significant sources of secondary organic aerosol, thus understanding
their formation pathways is relevant to atmospheric aerosol chemistry. This paper reports the success of PMF
factorization to differentiate HOMs originated from different sources in a boreal forest environment.





HOMs were measured with a CI-APi-TOF using nitrate ions for charging. Since the high-resolution peak-fitting may
introduce uncertainties that are not well quantified, we input unit-mass-resolution data as the data matrix, and identify
certain peaks with high-resolution afterwards. The error matrix is equally important to the data signal levels as an
input parameter in PMF. In this work, errors were estimated from laboratory data by fitting the statistical uncertainty
to the signal strength. The estimate shows good agreement with both that derived from an independent statistical
analysis of the ambient data, and also with the estimate widely used for aerosol mass spectrometrical data.

Mathematical diagnostics suggests that the error estimation is proper and that the model results are robust. At least
three factors are needed to explain most ($> 95\,\%$) of the observed spectral and temporal variations. In respect to the
interpretability, the data is optimally explained by six factors (accounting for 97% of the variability). Two nighttime
factors likely represent the oxidation of monoterpene initiated by $O_3$ and $NO_3$, respectively. The profile of the $O_3 +$
monoterpene factor is similar to the reference spectrum in previous chamber studies where only $O_3$ and monoterpenes
were injected, and the uncentred correlation coefficient between the factor and the reference spectrum is 0.91. The
$NO_3 +$ monoterpene reaction channel is supported by the detection of nitrogen-containing dimer compounds. In the
early morning, both nighttime chemistry channels are suppressed by NO reaction, shown by the appearance of factors
representing $RO_2 +$ NO reactions. The major peaks in the first daytime factor are $C_{10}H_{15}O_{6,8}NO_3$, whose parent $RO_2$
radicals are likely from $O_3 +$ monoterpene. Two other daytime factors are retrieved, though the underlying chemical
processes forming those components are not clearly understood. One daytime factor correlated well with sulfuric acid,
suggesting the chemistry represented by this factor could be controlled by the OH radical. The third daytime factor
contained many smaller HOM molecules and showed notable correlation with UVB and temperature. The
interpretation is that the formation of these smaller HOM molecules are OH-initiated, and their gas-phase
concentration is affected by temperature probably through particle-gas partitioning. Apart from these five "local"
factors, the sixth factor is interpreted as a transport factor, due to its similar temporal variation to $SO_2$ and its prominent
peak $C_6H_5NO_3$, a reported tracer of biomass burning.

Among the six factors retrieved by PMF, only the nighttime type-1 factor ($O_3 +$ monoterpene) has been confirmed in
the laboratory. However, the retrieval of this factor also strongly supports the validity of the model results. The
deduced chemical processes for the nighttime type-2 factor ($NO_3 +$ monoterpene) and the daytime type-1 factor ($RO_2$
+ NO) are supported by their correlations with other co-located measurements. To confirm and better understand these
two factors, laboratory experiments are needed to investigate the yields and dependence on other parameters. The
daytime factors are harder to interpret. However, testing the hypotheses we suggested based on PMF results will be a
good starting point for future studies. In summary, running PMF on CI-APi-TOF data was successful, and the results
presented in this paper improve our understanding of HOM production by confirming current knowledge and inspiring
future research directions.


*Acknowledgements.* Liine Heikkinen and Federico Bianchi are acknowledged for useful discussions. Mikhail
Paramonov, Jonathan Duplissy, Alessandro Franchin, Katrianne Lehtipalo, Hanna Manninen, Pasi Aalto, Juha



Kangasluoma, Emma Järvinen, Erik Herrmann and the personnel of the Hyytiälä forestry field station are
acknowledged for help during field measurements. This work was partially funded by Academy of Finland (1251427,
1139656, Finnish centre of excellence 1141135) and European Research Council (ATMNUCLE, grant 227463, and
COALA, grant 638703).

**Table 1**. Suggested elemental composition of fingerprint molecules of the six factors. *Peak fitting are shown in Fig.
S9.

| Factor | Fingerprint molecules |
|---|---|
| Nighttime type-1 | $C_{10}H_{14}O_7$, $C_{10}H_{15}O_8$, $C_{10}H_{14}O_9$, *$C_{10}H_{15}O_{10}$, $C_{20}H_{32}O_{11}$ |
| Nighttime type-2 | $C_{20}H_{31}O_8NO_3$, *$C_{20}H_{31}O_{10}NO_3$, $C_{20}H_{32}O_6(NO_3)_2$ |
| Daytime type-1 | *$C_{10}H_{15}O_6NO_3$, $C_{10}H_{15}O_8NO_3$ |
| Daytime type-2 | *$C_{10}H_{15}O_5NO_3$, $C_5H_6O_7$ |
| Daytime type-3 | *$C_3H_5O_3NO_3$, $C_4H_5O_4NO_3$, $C_5H_7O_4NO_3$, $C_6H_9O_4NO_3$, $C_7H_9O_4NO_3$ |
| Transport | *$C_6H_5NO_3$ |




**Table 2**. Suggested HOM formation pathways represented by each factor, and the correlation coefficient between
factors and other relevant conditions. In total, 1632 data points (30-min time resolution) are used. *These species
cannot be ruled out.

| Factors | Suggested main oxidant | Suggested main $RO_2$ terminator | correlation coefficient (R, n=1632) | | | |
|---|---|---|---|---|---|---|
| | | | NO | $H_2SO_4$ | UVB | T |
| Nighttime type-1 | $O_3$ | $RO_2$ | -0.26 | -0.32 | -0.18 | 0.22 |
| Nighttime type-2 | $NO_3$ | $RO_2$ | -0.23 | -0.32 | 0.04 | -0.13 |
| Daytime type-1 | $O_3$ | NO (*$HO_2$) | 0.56 | 0.32 | 0.16 | 0.40 |
| Daytime type-2 | OH (*$O_3$) | NO (*$HO_2$) | 0.09 | 0.77 | 0.53 | 0.65 |
| Daytime type-3 | OH (*$O_3$) | NO (*$HO_2$) | 0.17 | 0.48 | 0.68 | 0.36 |
| Transport factor | - | - | 0.35 | 0.01 | 0.09 | 0.12 |






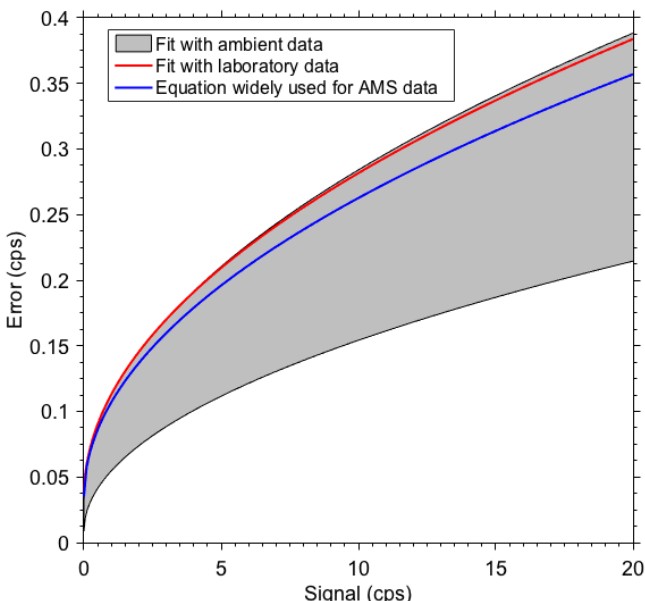


**Fig. 1.** Error matrix estimation by fitting the error to the signal strength. The red solid line is the best fitted curve from the laboratory experiment data with the customary fitting equation, the gray area represents the fitting from the ambient data with a different method (see supplementary information section S2), and the blue curve denotes the fitting equation commonly used for AMS data.



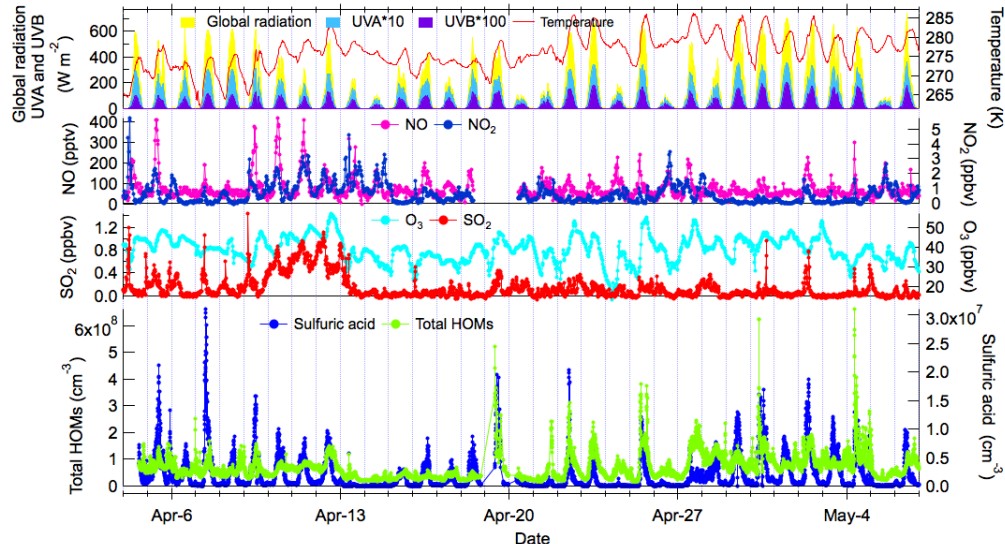

534

**Fig. 2**. Overview of the measurement from April 4 to May 8, 2012. The top panel shows meteorological parameters,
including UVA, UVB, global radiation, and temperature. Co-located measurements of inorganic trace gases, including
NO, NO$_2$, SO$_2$, and O$_3$ are shown in middle panels. Highly oxidized species measured by the CI-APi-TOF, i.e. sulfuric
acid (SA) and total HOMs, that are shown in the bottom panel.





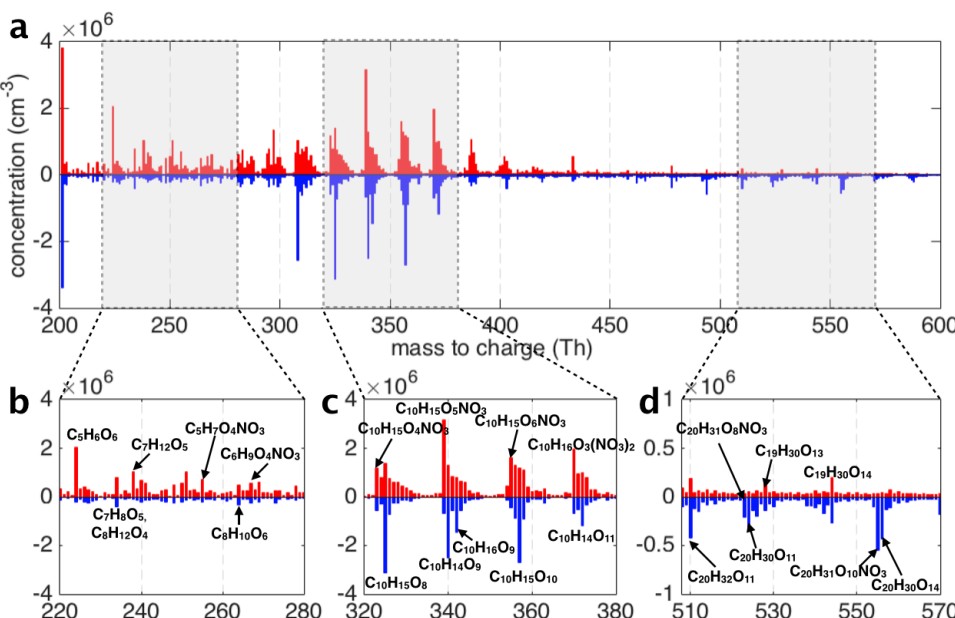


**Fig. 3**. Comparison of spectra measured by CI-APi-TOF between daytime and nighttime. The daytime spectrum
(marked in red) is above the zero line and the nighttime spectrum (marked in blue) is below the zero line. Fig. 3b, c,
and d present expanded mass spectra where major peaks are labeled with their possible elemental formula.



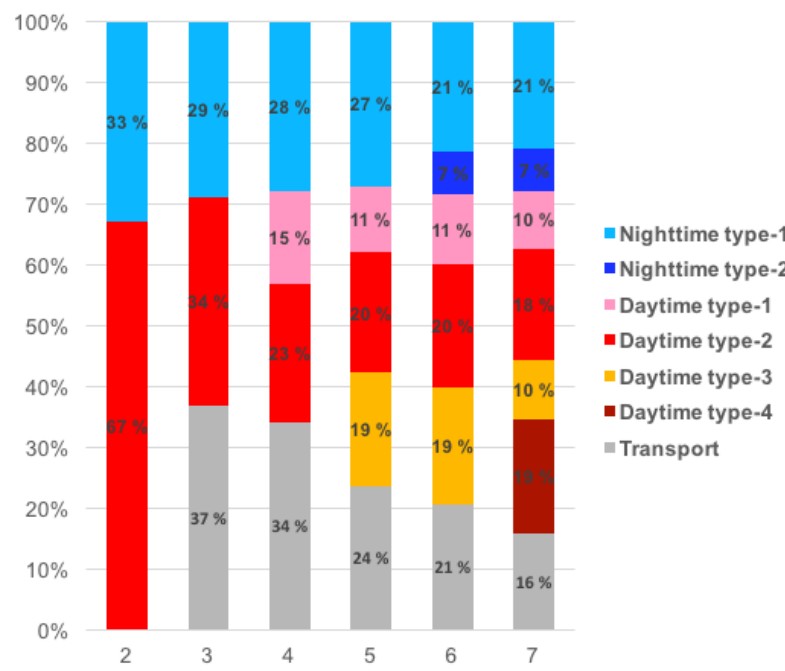


**Fig. 4**. Source allocation from 2-7 factor PMF solutions.



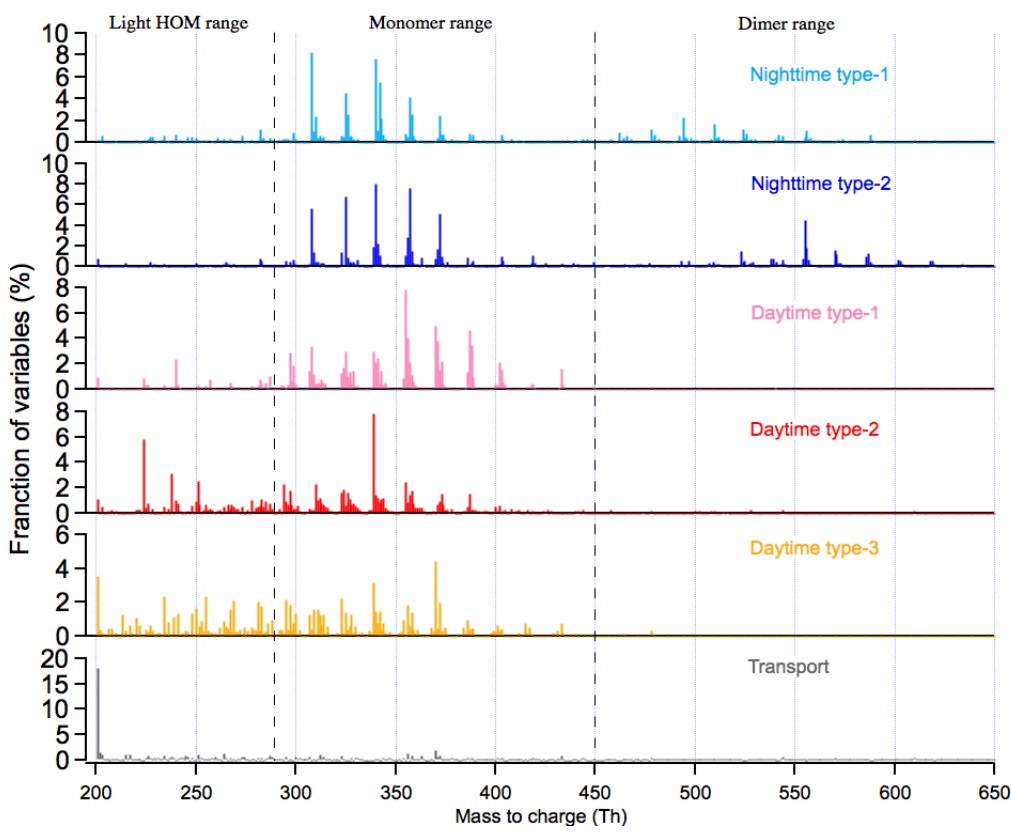


**Fig. 5**. Factor profiles in 6-factor PMF. The total signal of each factor is normalized to unity, and y-axis is the fraction of variables in the factor in percentage.

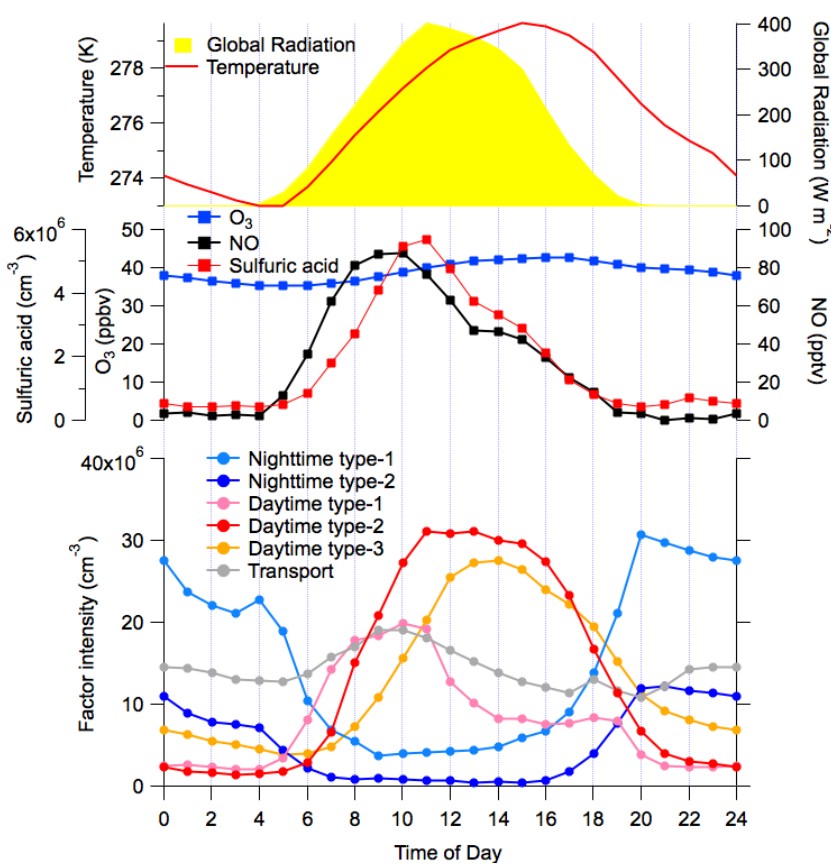


**Fig. 6**. The diurnal cycle of PMF factors, selected meteorological parameters, and trace gas concentration.





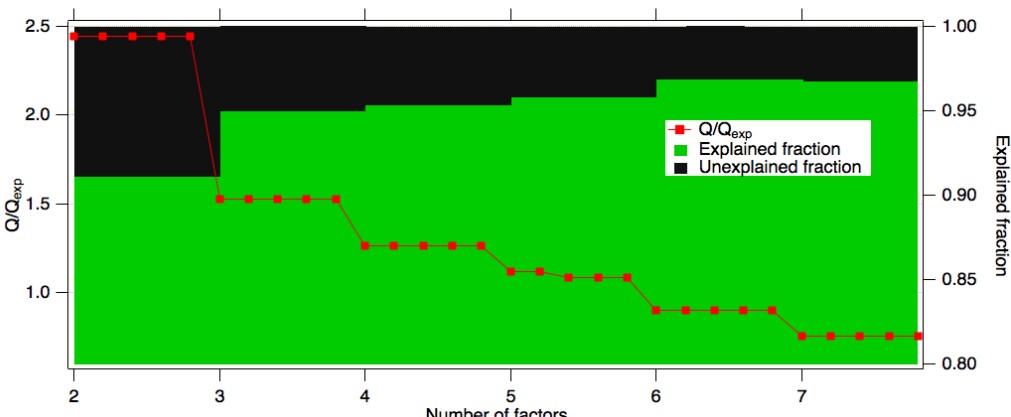


**Fig. 7**. Mathematical diagnostics of PMF solutions, including the overall changes of $Q/Q_{exp}$ and the explained variation
from 2-factor to 7-factor solutions. For each number of factors, five seed runs were performed to test the consistency
of the solution.





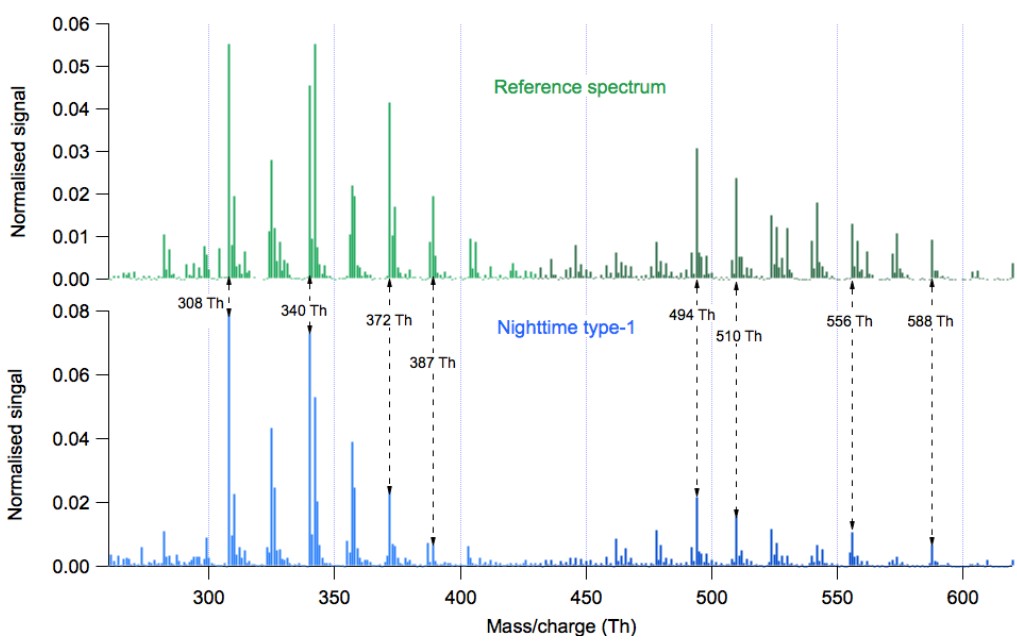


**Fig. 8**. Comparison between the reference spectrum (Ehn et al., 2014) and the O$_3$ + monoterpene factor.



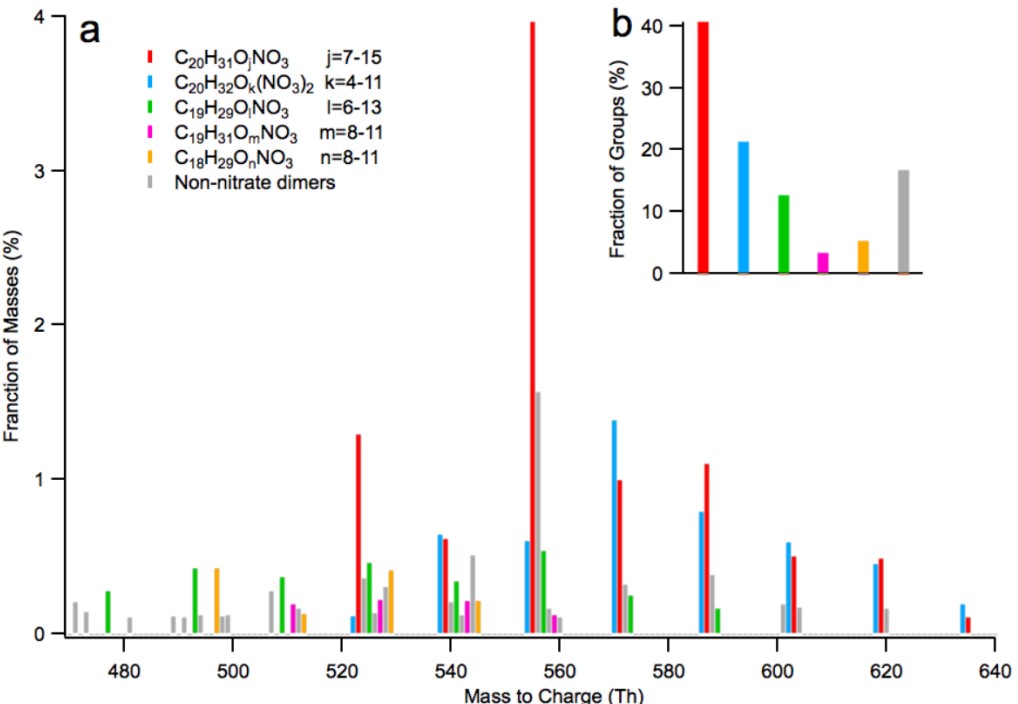


**Fig. 9**. Dimer profile of the nighttime type-2 factor. All dimer peaks are assigned to six groups based on their elemental
formula and marked with different colors. Fig. 9a shows the location and mass fraction of individual peaks, and Fig.
9b gives the fraction of these groups.



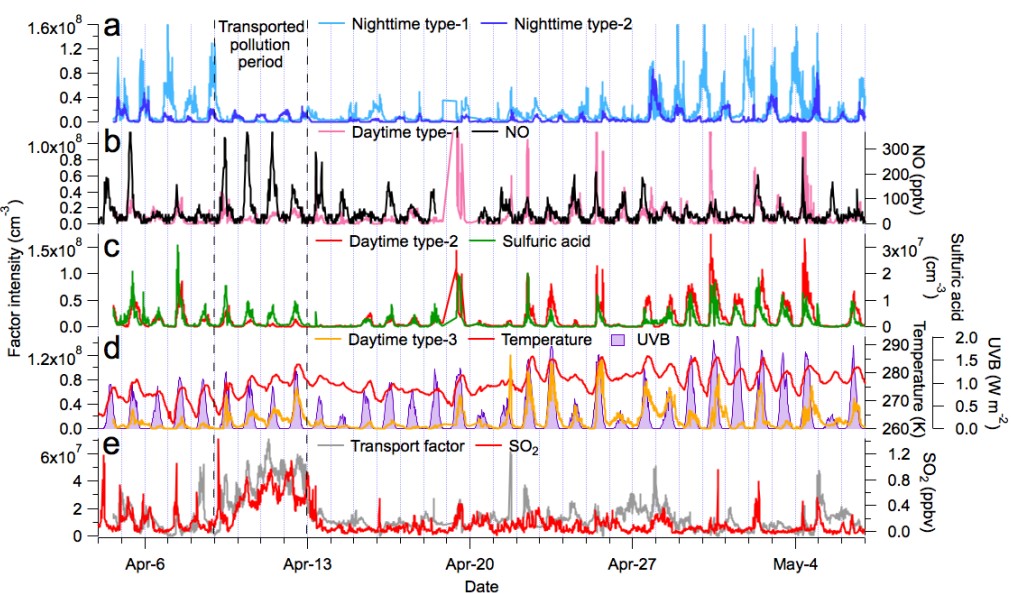


**Fig. 10**. Temporal behaviors of PMF factors and relevant tracer gases as well as meteorological conditions. The period
with transported pollution is marked by the dashed lines. Fig. 10a depicts the temporal variation of the two nighttime
factors. Fig. 10b shows the time series of Daytime type-1 factor together with NO. Fig. 10c demonstrates the similar
temporal behavior of the Daytime type-2 factor and sulfuric acid. Fig. 10d shows the time series of the Daytime type-
3 factor together with the relevant meteorological conditions (i.e. UVB and temperature). Fig. 10e depicts the temporal
variation of the transport factor, together with $SO_2$, a tracer for transported pollution.



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
