# Peer review of "Source characterization of Highly Oxidized Multifunctional 1 Compounds in a Boreal Forest Environment using Positive Matrix 2 Factorization"

_Atmospheric Chemistry and Physics, 2016_

## Referee Comment (RC1) · Anonymous Referee #1 · 9 Jun 2016

The authors analyse field observations of highly oxidized multifunctional molecules (HOM) by nitrate CIMS in Boreal forest (Hyytiälä). A period of more than 4 weeks was used to investigate in how far PMF would help to understand the (chemical) origin of different groups of HOMs. Critical points in PMF are the selection of the number of factors and appropriate treatment of errors. The authors find that three PMF factors are sufficient to catch major source signatures, however, 6 factors are more suited to describe evident finer details. The six factor solution is formally acceptable within the mathematical PMF control framework. The authors underline that formal criteria are not sufficient to judge PMF solutions and discuss the factor profile (MS) and factor time series in context of laboratory MS and the time series of other field observations.

The authors spent substantial efforts in determining the error matrix, which they try to describe in the supplement. The main manuscript is well structures and well written; figures and table were well selected. The manuscript is interesting to read. Unfortunately, the supplement is much weaker than the manuscript itself and suffers from typos and "looseness". By these and somewhat unclear notations it is difficult to follow it in large parts. This is unfortunate because a better edited supplement clearly could strengthen the whole manuscript. The value of PMF, especially in MS containing mainly molecular information (no fragmentation) is under debate. I think the results of this paper show that PMF applied in suited way to HR-CIMS can indeed help interpretation of field data. The manuscript should be published in ACP after a fes minor revisions. However, I urgently suggest to the authors to revisit their supplement and provide a better and clearer edition.

A general remark: I think the manuscript is very good and interesting. However, in parts you are using a relatively formal language. This is ok if you talk about general PMF. However, you are analyzing mass spectra. For a general reader (and me) it would be helpful if you would breakdown the general mathematical notations to the items you are de-facto dealing with: variables -> peak positions or m/z, analytical uncertainty -> standard deviation of instrumental noise, DL/3 -> 1sigma detection limit, DL -> 3sigma detection limit. etc. (see also below remarks to supplement).

Minor comments:

line 114: I think, only (X-H)- should appear in the numerator of eq. 1, as (HNO3)*(X-H)- is redundant and has the same mass as (NO3)*X-, and so on for i>0

line 167f: Let us assume overlap of two compounds, nearly the same mass and similar intensity arising in two different factors. How would you deal with separation of the right compound into the right factor? Isn't the argument, that you do not have much of such overlap? Therefore you can use UMR and determine the according elemental composition later. Did you apply any diagnostics to show that overlapping peaks of

minor importance?

line 179: How would you define contributions to analytical uncertainty? Is it instrumental noise, is it background, is it interferences at or near a given m/z? I think the manuscript and even more the supplement would become clearer if with a more consistent and traditional notation.

line 199: Where does the value DL = 0.105 come from? I guess it is no accidence that it is 3 x "the background for all tunings in the 'blank mass' (800~1000 Th)" "estimated to be 0.035". I think this is confusing, see previous remark..

line 388ff: How does the dimer analysis of factor type -1 compares to factor type-2 in Figure 9 ? Do you find nitrate containing dimers?

line 404ff: Could one state that daytime type 1 factors is of special (relative) importance when UVB and OH are low, thus daytime ozonlysis gains in importance?

line 413 and 434: Plants emit more than monoterpenes (even toluene): could isoprene, methyl salicylate, or so called leaf alcohols play a role?

line 433f: "could" maybe better than "may"

line 456: "in principle" maybe work better as "theoretically. No theory in involved, only expectations.

line 461 and Table 2: Are these uncentered correlation (UC) coefficients as announced in line 333 ? Please clarify.

line 477: You are revealing the chemical sources with exception of the transport factor. Is it possible to check for correlations with monoterpene emissions or so? Could that help for the non-C10 observations?

Figure 1 and Supplement Line 95 + 112f: Eq. S1 should be also plotted in Figure 1. Otherwise I cannot understand why there are lines for AMS and lab results, but a range for ambient data approach. More important, as I understood you expect to determine

the upper limit of your error by analysis of the ambient data. This was inherent to the method applied, as you could not exclude real chemical variations within the analysis interval. Why is the lab approach almost at the top error boundary of the approach using ambient data. Isn't that a contradiction?

typo and errors line 101: Eastern line 254: Mix of singular and plural line 419: Tröstl et al is missing in the reference list

SUPPLEMENT

line 5: Why is ion detection by HR-CIMS more complicated the by AMS? I think it is easier because of limited fragmentation.

line 8: How does transmission affect signal background ? Because of real signals arising from "contaminations"?

line 9: No a-value in eq. 6!

line 12: Mix of singular and plural.

line 14: mlpm in manuscript

line 15: "without generating large turbulence" I doubt that looking at your set up in Figure 1. Moreover you want turbulence to mix your calibrant with the main flow.

line 16: This last sentence does not make really sense: "vacuum line"?

line 20: "used IN the permeation source"

line 26: Eq. 5 ?

line 26f: I understand background as "offset". But you are looking at instrumental noise?! The background, I would determine around each m/z = 0.5UMR, i.e. between two peaks. I also would try to determine the instrument noise there.

line 31 and Fig. S3: I think, there is a trend of increasing "back ground" with decreasing m/z. 0.035 is background or instrumental noise, or detection limit ? See my previous

comment.

line 35: eq 6 ??

line 51+ 54: "median over a short period of time (5 data points)"; does that mean over 25 min.? Then you might have indeed to consider influence by chemical changes!?

line 55f: I really don't understand what you did. Especially the last half sentence is unclear. Try to reformulate in clearer way.

line 64f: "dividing the "noise estimate" (i.e. signal minus trend) data into bins"; difficult to understand.

line 69: S1-S9 (?), but you need 10 bins! From here on, you mix the notation "S1- S9" and the fact that your using 10 bins. Check text and figures for that and correct.

line 73: 940? or 9084/10??

---

## Referee Comment (RC2) · P. Paatero (Referee) · 20 Jun 2016

This manuscript describes PMF analysis of a large matrix of time-of-flight spectra of ions formed of atmospheric VOC molecules. The main part of the ms is interpretation of the obtained six factors. This main part of the work is not discussed in this review.

The ms puts much emphasis in deriving reliable uncertainty estimates for the elements of the measured mass spectra. This is intended in order to provide firm foundation for PMF modeling of these measurements, also for analysis of future similar measurements.

The comparison of obtained uncertainty estimates with obtained residual values is

badly erroneous. Hence, conclusions about quality of fit are also erroneous. I recommend that this manuscript should be published in ACP after this comparison is performed correctly and all text based on this comparison is rewritten according to the corrected comparison. Also, I request that all the numerous problems discussed below are corrected (or present text is enhanced so that correctness of present text becomes evident).

The abstract says

PMF was performed with a revised error estimation derived from laboratory data, and this approach was validated by mathematical diagnostics of the PMF solutions.

Unfortunately, this statement is erroneous. Mathematical diagnostics indicate that the carefully derived uncertainty estimates of data values are in striking conflict with the residuals obtained by PMF modeling.

The main part of this review is concerned with this discrepancy. Additionally, here are remarks regarding different erroneous or questionable details in the presentation. Although I will present some criticism regarding the estimation of uncertainty estimates, I believe that these estimates are sufficiently accurate so that when the computed residuals are seen to be much larger than the estimated uncertainties, then the discrepancy is real, not caused by errors in uncertainty estimates.

Essential mathematical diagnostics are only found in the Supplement. No hint of them is found in the main text.

Section 3 of Supplement is "Examining Q distribution of time and variables" It is good that this data is presented. However, its interpretation was not right, as shown in the following.

IT IS ASSUMED THAT MODEL ASSUMPTIONS OF PMF DO HOLD

If model assumptions do hold, then residuals are only due to data noise, so that assumed data uncertainties agree with observed distributions of residuals.

If model assumptions do hold, then all profiles stay unchanged throughout the measurement period, and the assumed number of factors is right.

Notation: The dimensions of the matrix are m rows, n columns. There are p factors. Q sums over columns and rows are denoted by $Q\_j$ and $Q\_i$, respectively. When we say "residuals" we mean scaled residuals, i.e. residuals divided by respective assumed data uncertainties.

THEORETICAL VARIATION OF Qrow AND Qcolumn VALUES

The Supplement says:

"The mean value of Q on all variables was well below 4, the threshold in robust-mode PMF. This suggests that all variables are well described by the model."

These sentences confuses single point $Q\_{ij}$ contributions with the overall Q sums obtained for an entire matrix, for an entire row, or for an entire column. It is possible (within model assumptions) that a few individual points get residuals >4, whereby the $Q\_{ij}$ contributions from such points exceed 16.

Estimates for Q contributions due to columns or rows are obtained from Statistical theory. Approximately, theory says that the expected value of $Q\_j$ from any column j is = m-p, and from any row i, $Q\_i$ = n-p. It also says that approximately the statistical distribution of Q is equal to chi-squared distribution whose degrees of freedom is m-p for $Q\_j$ and n-p for $Q\_i$.

In the present work, p«n and p«m, thus we may approximate: Distributions are: chi2(n) for row Q's and chi2(m) for column Q's. As m and n are large, chi2 is well approximated by the normal distribution. Thus distributions of Q values are approximately:

for row Q's, distribution of $Q\_i$ is N(n,sqrt(n))

for column Q's, distribution of $Q\_j$ is N(m,sqrt(m))

where N(*,*) denotes normal distibution and sqrt(m) and sqrt(n) are the standard deviations of respective normal distributions. As an example, compute limits of these distributions for n=400, m=10000. Then the lower and upper 2-sigma limits for Q values, under model assumptions, are

360 to 440 for row Q's

9800 to 10200 for column Q's

It is seen that Q values come very close to their expected values m and n when model assumption are valid. Such "well-behaving" Q values are obtained in numerical simulations when the only simulated error is the random error in data values. If computed Q values deviate more in analyses of real data, then random noise in data values cannot be the explanation if assumed uncertainties are correct for data noise.

COMPARISONS WITH Q VALUES SHOWN IN SUPPLEMENT

There appears to be a conflict between parts a and b of Fig. S8. The overall averages of Q in parts a and b should be identical because they are averages of the same matrix of individual $Q_{ij}$ values. In the following discussion, we only consider part b of Fig. S8. which seems to agree with the value of Q/Qexp for 6 factors that is given for the overall Q in Fig 7 of the manuscript proper.

Distribution of Qrow values (fig S8/b) ranges from approximately 0.2n to 2n, some rows even exceed these limits. This variation is very much wider than the expected width of chi2 distribution for Qrow.

Thus it is concluded that model assumptions did not hold for this PMF modeling. The estimated noise in measured values does not explain the observed variation of Qrow from row to row. Small values of Qrow may have a simple explanation: censoring of values <DL (see below) has eliminated some variation from the data, so that Q contributions from BDL values are much less than expected. Thus Qrow values of low-intensity rows will be (much) smaller than expected.

Note that downweighting low-intensity ("weak" and "bad") columns will also decrease

both Qrow and Qcolumn values below their expected values.

On the other hand, there are significant numbers of rows with

Qrow > n + 4 sqrt(n).

It is difficult to know the reasons for these large Qrow values. Possible reasons are e.g.:

- variation of component profiles from sample to sample.

- small slow variation of mass calibration and/or resolution from sample to sample.

- small variation of critical parameters of the ionization process, e.g. of temperatures

Careful study of residuals would be the first step in finding the reason(s) for increased Qrow and Qcol values.

It is essential to admit that "something" happens in the atmosphere, or in the measurement process, that is not currently understood.

One must not try to "explain away" this "something" by arguing that yes, this was already seen by others, there is nothing new in such variation of row and column Q values.

On the contrary, this is indeed not an exceptional case. It is a phenomenon that should be studied so that it is understood. If there is component profile variation, understanding this variation might be a significant step in understanding chemical processes in the atmosphere.

It is important to modify the manuscript so that this "something" is clearly presented. It is not necessary nor possible to determine in this manuscript the reasons behind the observed Q variation.

It might perhaps be good to discuss or mention possible reasons. Figure S8 is crucial in demonstrating this Q variation. It might be good to move Fig S8 to the main text.

After all, determination of data uncertainties is one of the main contents of the paper. Importance of Fig S8 is based on carefully determined data uncertainties, thus it is not logical to hide Fig S8 in the Supplement.

It would be good to point out that the observed good overall Qexp values are misleading in this case. Some Qrow values are much too large while others are much too small, so that these two effects largely cancel each other in the value of overall Qexp. See also remark re lines 306,307, below.

DETAILED DISCUSSION OF THE MAIN PART

Eq(1) is unclear. What quantities are represented by [X] and by the numerator and denominator. The text says

"the numerator on the right hand side is the sum of all detected ions"

This probably means: ... is the sum of detected ion concentrations?

Further: "the denominator is the sum of all reagent ion signals" What does this mean? Note that there are no square brackets in the denominator.

On lines 126, 132, 176, and possibly elsewhere, PMF is called "an algorithm". This is wrong. PMF is a model, it defines the equations that should be fulfilled by the computed factor elements. Algorithm is a procedure for finding the values for factor elements so that they fulfill the model. There are currently at least 4 different algorithms for fitting or "solving" this model PMF. Please use correct terminology! Admittedly, the majority of chemically oriented papers do not pay attention to this distinction. In fact, it would be good to specify which PMF algorithm was used: the original algorithm in PMF2, or a PMF script executed by program ME-2? There are slight differences between these programs, especially if there is rotational ambiguity in the model. (There is probably very little rotational ambiguity in this work, so that the distinction PMF2 vs. ME-2 does not matter now. However, it is good manners to specify the used tools.)

Lines 167 - 169 say "... Therefore, the data matrix used in this work is in unit-mass

resolution, and peak fitting was performed afterwards to identify the elemental formula of peaks..."

This is an important decision and probably quite suitable for these data sets. There would also be other ways of formulating the matrix. It might be useful to learn whether the authors experimented with different ways, and what considerations lead them to select the unit-mass resolution. However, if the authors plan to examine this question later in more detail in another paper, then it is oK to not discuss this question now.

Lines 184-186 say " $I$ is the signal strength (ions/second) of the ion, $t_s$ is the integration time in seconds, and $a$ is a factor accounting for the fact that a single ion will generate a Gaussian-shaped pulse in the detector, rather than a single peak. "

Here is confusion (or sloppy wording). I am not sure how to understand this topic. First, I believe that the words

"generate a Gaussian-shaped pulse in the detector"

are a mistake. The intention is probably to say that

"individual ions produce pulses whose pulse height distribution is of Gaussian shape."

Second, why does the pulse height distribution matter at all? If ions are not actually counted but count rate is determined by integrating the current that is due to accumulated pulses, then the statement would be understandable: the variation of charge produced by each ion does indeed contribute to the uncertainty of integrated current. In contrast, if ion pulses are actually counted (I believe this is the case), then the variation of pulse height from ion to ion does not directly contribute to uncertainty, except if the variation is large enough so that a fraction of ions are not counted at all. Please clarify or correct.

Section 2.3.2. It would be good to state clearly that the data matrix consists of counts-per-second values, obtained as 5-minute averages. This fact can be inferred from the present text but why not help the reader by stating it explicitly.

The paragraph beginning on line 209 claims that Paatero et al (2003) recommended censoring variables that are below DL by fixing the values at DL/3 and uncertainty at DL. This claim is entirely fictitious and wrong. There is not a single word suggesting censoring BDL values in the 2003 paper.

In contrast, certain PMF-related papers advise against this practice. There is no demonstrated benefit from this practice, provided that low S/N (i.e. "weak" and "bad") variables (entire columns) are downweighted as recommended in that 2003 paper. On the other hand, my personal experience in reviewing has revealed several cases where such censoring created one or two ghost factors, i.e. numerical artefacts caused by censoring. Also, censoring often creates bias in the results.

It is possible (likely?) that in the present case, censoring BDL values did not cause noticeable harm in main results because there were so many strong variables. The only likely harm might be that some details were lost from those columns where itensities are lowest. Thus it is not reasonable to suggest that the work should be redone using the original measured BDL values and uncertainties. On the other hand, the present formulation of the paper would be interpreted by your readers as a rule saying that BDL values -must- be censored. In order to help prudent practices prevail among atmospheric scientists, I request the following addition in the ms:

After explaining that BDL values were replaced by DL/3 in this work, you shall insert a remark, something like the following:

After this work was completed, we became aware that this practice of replacing BDL values by fixed values is harmful and provides no advantage at all, although in this specific case, the main results were possibly not harmed. Thus we emphasize that in future studies, our example should -not- be followed. Instead, values < DL and their uncertainties should remain unchanged in data and error matrices.

Signal-to-noise estimates (S/N) are discussed in the paragraph beginning on line 209. Please state which formulation of S/N was used. There are two published formulations:

(1) the recommended S/N definition, distributed with EPA PMF v5 and published in the Supplement of

Brown, S. G., Eberly, S., Paatero, P. & Norris, G. A., Methods for estimating uncertainty in PMF solutions: Examples with ambient air and water quality data and guidance on reporting PMF results. Science of the Total Environment. 518-519, p. 626-635, 2015

(2) the earlier problematic S/N definition, suggested in the quoted 2003 paper and used in all earlier EPA PMF versions.

The numerical values produced by these two methods differ from each other, thus the readers need to know which method was used by you: one of these, or your own method (define).

The manuscript mentions that some columns were downweighted (DW) by 2 or by 10. State how many columns were DW by 2 and by 10. Numbers of DW columns also influence the Qexp values. When you reported Q/Qexp, did you use correct Qexp values that take this influence into account? If not, state this clearly! If yes, state that, too!

Lines 306, 307 say "From two to seven factors, Q/Qexp decreases stepwise from 2.44 to 0.76. The closeness to unity indicates that the estimated error is appropriate for the model."

This good agreement of Qexp with its theoretical value is misleading (see above, too). Because of downweighting and/or censoring, low concentrations contributed to Qexp much less than expected. This is OK once it is recognized. However, there are also high concentration values that contribute to Qexp much more than expected, so that the overall Qexp appears acceptable. Thus it is not right to claim that the estimated error is appropriate for this PMF modeling with 6 or 7 factors.

DETAILED DISCUSSION OF THE SUPPLEMENT

There are too many typos and broken sentences, poor language, and even mistakes

in the equations. It must be emphasized that all text, even the Supplement, should be carefully checked by one or two of senior authors before the manuscript is submitted.

Language must be improved in the whole of Supplement text. This is essential in order that the text may be understood!

lines 9, 10, 35

These lines refer to "a" in Eq(6). There is no "a" in Eq(6)? Confusion? Is Eq(7) intended? This confusion may be present elsewhere, too

line 33 says

"also confirms the validity of the pre-assumption" which pre-assumption? Do not pose extra difficulties for the reader, please be explicit!

line 64 says

"by dividing the "noise estimate" (i.e. signal minus trend) data" what does "trend" mean here. I cannot even guess.

lines 64 to 70

This paragraph is almost impossible to understand. Equations are the language of mathematics. Please use equations as the main method of defining what was done. Verbal explanations may only be used as a help for understanding the equations.

lines 82 to 89

This paragraph confuses superposition and convolution. First do the math properly, then rewrite the paragraph.

Fig S1, caption contains: "All the flows were set identical throughout the experiments"

better to write: "All the flows were kept unchanged throughout the experiments"

or "All the flows were constant throughout the experiments"

Eq(S1)

This equation contains a parameter "a". The "Allan equation" Eq(7) in the main text also contains a parameter "a". However, the equations are different, and the "a" values are thus different, too. In Eq(7), the "a" is dimensionless and approximately =1. In Eq(S1), the "a" has dimension and its value depends on integration time.

This confuses the reader significantly and quite unnecessarily. Please rewrite the supplement so that the same equation is used in both texts.

lines 103-104 say

"Parameter "a" is similar the "a" in the Allan et al. (2003) equation".

This statement is badly misleading, as already noted, above. If Eq (S1) is not changed as I suggested, then this statement must be changed to its opposite, warning the reader that the two symbols "a" are not the same.

Eq(S2)

There is a problem with this equation. I suspect that an equals sign is missing before the square root.

Eq(S3)

I do not understand this equation at all. What are the X symbols? What is the equation trying to say?

Fig S7.

These confidence limits for sigma values do not make sense. There must be some problem in their evaluation. Possibly, an invalid estimation principle was used.

Fig S8

I assume that s8/a represents the 6-factor solution (why is it not stated?). Then Figs a and b are based on same Q_ij values. However S8/b and S8/a of Q/Qexp seem to

be in conflict: there are many more values >1 in S8/b than in S8/a. Is there a natural explanation (show the explanation if there is one) or is there an error in generating the figures?

Fig S9

The caption says "... the purple one is the residue ..."

The correct term in numerical context is "residual". In chemistry, "residue" might be used for remains of substances.

---

## Author Comment (AC1) · 16 Aug 2016

**Comments from referees are in black, and our responses are in green.**

**Response to the comments of referee 1:**

General Comments

The authors analyse field observations of highly oxidized multifunctional molecules (HOM) by nitrate CIMS in Boreal forest (Hyytiälä). A period of more than 4 weeks was used to investigate in how far PMF would help to understand the (chemical) origin of different groups of HOMs. Critical points in PMF are the selection of the number of factors and appropriate treatment of errors. The authors find that three PMF factors are sufficient to catch major source signatures, however, 6 factors are more suited to describe evident finer details. The six factor solution is formally acceptable within the mathematical PMF control framework. The authors underline that formal criteria are not sufficient to judge PMF solutions and discuss the factor profile (MS) and factor time series in context of laboratory MS and the time series of other field observations.

The authors spent substantial efforts in determining the error matrix, which they try to describe in the supplement. The main manuscript is well structures and well written; figures and table were well selected. The manuscript is interesting to read. Unfortunately, the supplement is much weaker than the manuscript itself and suffers from typos and "looseness". By these and somewhat unclear notations it is difficult to follow it in large parts. This is unfortunate because a better edited supplement clearly could strengthen the whole manuscript. The value of PMF, especially in MS containing mainly molecular information (no fragmentation) is under debate. I think the results of this paper show that PMF applied in suited way to HR-CIMS can indeed help interpretation of field data. The manuscript should be published in ACP after a few minor revisions. However, I urgently suggest to the authors to revisit their supplement and provide a better and clearer edition.

A general remark: I think the manuscript is very good and interesting. However, in parts you are using a relatively formal language. This is ok if you talk about general PMF. However, you are analyzing mass spectra. For a general reader (and me) it would be helpful if you would breakdown the general mathematical notations to the items you are de-facto dealing with: variables -> peak positions or m/z, analytical uncertainty -> standard deviation of instrumental noise, DL/3 -> 1sigma detection limit, DL -> 3sigma detection limit. etc. (see also below remarks to supplement).

We would like to thank the referee for the helpful and detailed comments and suggestions.

According to the comments, we modified the manuscript in the following aspects:
1) Improve the supplement to make it more clear and easy to follow;
2) Replace the formal PMF language by mass spectrometry-oriented terminology to make the manuscript easily followed by general readers, such as Eq.2, Eq.6, and corresponding notations in the supplement.

In the following, we reply to the referee's comments item by item.

Minor comments:

line 114: I think, only (X-H)- should appear in the numerator of eq. 1, as (HNO3)*(X-H)- is redundant and has the same mass as (NO3)*X-, and so on for i>0

This equation is the same as used by Ehn et al. (2014). We used this equation here to emphasize our awareness of structurally different clusters with the same elemental composition, such as $(HNO_3)(X-H)^-$ and $X(NO_3^-)$. In practice, we are unable to distinguish them. We will simplify this equation as below:

$$[HOM] = \frac{HOM(NO_3^-)}{\sum_{i=0}^{2}(HNO_3)_i(NO_3^-)} \times C$$

We also omit the cases of $HOM(HNO_3)NO_3^-$ and $HOM(HNO_3)_2NO_3^-$, as in our analysis, these clusters' contribution are minor.

line 167f: Let us assume overlap of two compounds, nearly the same mass and similar intensity arising in two different factors. How would you deal with separation of the right compound into the right factor? Isn't the argument, that you do not have much of such overlap? Therefore you can use UMR and determine the according elemental composition later. Did you apply any diagnostics to show that overlapping peaks of minor importance?

In principle, PMF should be able to correctly attribute that (unit m/z) signal to two factors, with ratio equal to their actual contribution to the total signal at that m/z. The same issue can occur also in HR data, if a certain molecule has two different sources, then properly configured PMF will separate the signal from that molecule into two different factors. This is, in fact, one of the strengths of PMF.

Peak overlap on one unit-mass was indeed observed in our spectrum, but we did not perform any diagnostics analysis to show the potential effect of peak overlap, since we think this method should not lead to large uncertainties in our case for following reasons:

1) when interpreting factors, we often rely on fingerprint molecules and the most evident difference between factors is whether their fingerprint peaks contain nitrogen atoms, which are easy to separate simply from their masses (odd or even masses according to the "nitrogen rule", assuming closed shell products).

2) the most prominent peaks in our spectra are almost always dominated by one peak (usually > 90 %). In such cases, ignoring the minor peaks, which are more likely to be assigned correctly, probably would not lead to large uncertainties. When we determine fingerprint molecules for each factors, we only choose single or overwhelming peaks (e.g. Table 1 and Fig. S9).

3) in some cases, overlapped peaks are very likely from the same formation pathway so that we don't need to consider their separation by PMF. One example is that $C_{20}H_{32}O_{10}$ and $C_{19}H_{28}O_{11}$ overlap on mass 494 amu, but they are both dimer compounds likely from $RO_2$-$RO_2$ reaction, and indeed, PMF attributes the entire signal at 494 amu into the same factor.

line 179: How would you define contributions to analytical uncertainty? Is it instrumental noise, is it background, is it interferences at or near a given m/z? I think the manuscript and

even more the supplement would become clearer if with a more consistent and traditional notation.

The analytical uncertainty ($\sigma_{ij}$) is from counting statistics. As the number of available molecules is high, but the probability of ionizing and detecting such molecules is low, we can assume that the probable distribution of detected ion numbers for a given molecule population can be modeled as a Poisson distribution (Allan et al. 2003).

Allan, J. D., Jimenez, J. L., Williams, P. I., Alfarra, M. R., Bower, K. N., Jayne, J. T., Coe, H., and Worsnop, D. R.: Quantitative sampling using an Aerodyne aerosol mass spectrometer 1. Techniques of data interpretation and error analysis, J. Geophys. Res., C: Oceans Atmos., 108, 2003.

line 199: Where does the value DL = 0.105 come from? I guess it is no accidence that it is 3 x "the background for all tunings in the 'blank mass' (800~1000 Th)" "estimated to be 0.035". I think this is confusing, see previous remark.

We had some confusing terminology in the supplement. The background determined from "blank mass" is the standard deviation ($1\sigma$), and we define DL as $3\sigma$, that is why DL=3*0.035. We will change the terminology in both the main text and the supplement. Such as in Eq.6, the DL/3 is replaced by $\sigma_{noise}$, which is more straightforward.

line 388ff: How does the dimer analysis of factor type -1 compares to factor type-2 in Figure 9? Do you find nitrate containing dimers?

Yes, we have also checked the dimer distribution of nighttime type-1 factor, as shown in Figure 1. Though we see some mononitrate dimers in this factor, their contribution is small (16 %) compared with non-nitrate dimer compounds (84 %); the dinitrate dimers are not present in this factor. We think this is reasonable: these mononitrate dimers most likely have one parent $RO_2$ from $O_3$ + monoterpene reaction, so their temporal variation should also be affected by $O_3$ + monoterpene (type -1 factor); the parent $RO_2$'s of dinitrate dimer are both from $NO_3$ + monoterpene, which has no dependence on $O_3$ + monoterpene, so they are not present in type -1 factor. We will mention this in the main text.

[Figure]

**Fig. 1**. Dimer profile of the nighttime type-1 factor. All dimer peaks are assigned to six groups based on their elemental formula and marked with different colors. Fig. 1a shows the location and mass fraction of individual peaks, and Fig. 1b gives the fraction of these groups.

line 404ff: Could one state that daytime type 1 factors is of special (relative) importance when UVB and OH are low, thus daytime ozonlysis gains in importance?

The reviewer is correct that, in general, daytime ozonolysis will be the dominant loss pathway for monoterpenes during overcast days, as the OH levels stay low compared to sunny days. However, as so many other important parameters are also affected to a larger or smaller extent (e.g. ozone production, temperature mixing height, monoterpene emissions) we prefer to not discuss the possible effects following meteorological changes further than stating the suggested main oxidants in Table 2 in the paper.

line 413 and 434: Plants emit more than monoterpenes (even toluene): could isoprene, methyl salicylate, or so called leaf alcohols play a role?

We cannot rule out the potential contribution of other compounds, but the predominance of molecules with ten C atoms in them for each factor, together with good agreement with reference monoterpene spectra for most factors, suggests that monoterpenes are the main contributor. For isoprene, its concentration is low at our measurement site as measured with PTR-CIMS, and its oxidation products, such as $C_5H_{10}O_x$, are only minor signals in our spectra.

line 433f: "could" maybe better than "may"

Agreed, we will change it to "could".

line 456: "in principle" maybe work better as "theoretically. No theory in involved, only expectations.

Agreed, we will change it to "in principle".

line 461 and Table 2: Are these uncentered correlation (UC) coefficients as announced in line 333? Please clarify.

These were not UC coefficient, but Pearson's linear correlation coefficient. We changed to UC coefficient. The two types of coefficients give a similar picture.

Table 2. Suggested HOM formation pathways represented by each factor, and the uncentered correlation coefficient between factors and other relevant conditions. In total, 1491 data points

| Factors | Suggested main oxidant | Suggested main $RO_2$ terminator | correlation coefficient (R, n=1632) | | | |
|---|---|---|---|---|---|---|
| | | | NO | $H_2SO_4$ | UVB | T |
| Nighttime type-1 | $O_3$ | $RO_2$ | -0.05 | 0.19 | 0.14 | 0.26 |
| Nighttime type-2 | $NO_3$ | $RO_2$ | -0.07 | 0.18 | 0.06 | 0.33 |
| Daytime type-1 | $O_3$ | NO (*$HO_2$) | **0.38** | 0.56 | 0.50 | 0.42 |
| Daytime type-2 | OH (*$O_3$) | NO (*$HO_2$) | 0.22 | **0.76** | **0.84** | 0.69 |
| Daytime type-3 | OH (*$O_3$) | NO (*$HO_2$) | 0.29 | 0.56 | 0.66 | **0.80** |
| Transport factor | - | - | 0.41 | 0.48 | 0.44 | 0.51 |

(30-min time resolution) are used. *These species cannot be ruled out.

line 477: You are revealing the chemical sources with exception of the transport factor. Is it possible to check for correlations with monoterpene emissions or so? Could that help for the non-C10 observations?

While some monoterpene emission measurements are available, these are not easily up scalable to regional scale that would be required here. But in any case, the actual monoterpene concentrations would be the more useful comparison. Unfortunately, the data quality of monoterpenes measured by PTRMS was not very good for this period. However, in general, the diurnal pattern of monoterpene shows low concentration in the daytime and elevated concentration during the night, driven by variations in mixing height. This trend is similar to our nighttime factors but opposite to all daytime factors, even though we are confident (e.g. based on chamber studies) that the main daytime HOM signals are from monoterpene oxidation.

In other words, while it is very possible that other VOCs than monoterpenes contribute to the HOM spectra, it is unfortunately not easy to draw any conclusions on this from the measured VOC concentrations we have available.

Figure 1 and Supplement Line 95 + 112f: Eq. S1 should be also plotted in Figure 1. Otherwise I cannot understand why there are lines for AMS and lab results, but a range for ambient data approach.

Agreed. We now revised the equation for this ambient data approach, and plot it in Figure 1.

More important, as I understood you expect to determine the upper limit of your error by analysis of the ambient data. This was inherent to the method applied, as you could not exclude real chemical variations within the analysis interval. Why is the lab approach almost at the top error boundary of the approach using ambient data. Isn't that a contradiction?

The lab-derived estimate and the error estimated from ambient data are estimated from two independent statistical methods, as described in the text, and our primary aim was to see if the two estimates roughly match each other. The shown agreement in Figure 1 is quite good considering this inherent difference, and should not be considered in contradiction.

typo and errors line 101: Eastern line 254: Mix of singular and plural

We now corrected these typo and errors.

line 419: Tröstl et al is missing in the reference list

We added Tröstl et al (2016) to the reference list.

SUPPLEMENT

line 5: Why is ion detection by HR-CIMS more complicated the by AMS? I think it is easier because of limited fragmentation.

Our sentence was confusing. What we want to emphasize is that CI-APi-TOF is usually set to detect ions on a much broader mass range, over which the detection efficiency (transmission) might change significantly. We need to examine if the transmission affects the counting uncertainty. We will remove this sentence.

line 8: How does transmission affect signal background? Because of real signals arising from "contaminations"?

This was indeed improperly phrased. What we wanted to emphasize is that whether the transmission will influence the signal counting uncertainty is unknown and needs examination. We will modify it as "Its potential influence on signal counting statistics needs examination."

line 9: No a-value in eq. 6!

It should be eq.7 instead of eq.6. We will correct it.

line 12: Mix of singular and plural.

We rephrase the sentence as "A temperature controlled permeation source was connected to the CI-inlet"

line 14: mlpm in manuscript

This is a typo. It should be milliliter per minute (mlpm) in the supplement.

line 15: "without generating large turbulence" I doubt that looking at your set up in Figure 1.

Moreover you want turburlence to mix your calibrant with the main flow.

This is the same typo as in line14, the unit "slpm" should be "mlpm". We will correct this typo in the figure.

line 16: This last sentence does not make really sense: "vacuum line"?

We rephrased the sentence in line 16 as "The outflow of the permeation source was further diluted by $N_2$ flow (~ 10 lpm) before entering the chemical ionization inlet (CI-inlet) as the sample flow. The flow rate of the sample flow could be adjusted by varying the total flow and/or sheath flow of the CI-inlet, which were set to 30 lpm and 20 lpm, respectively. All these flow rates were kept identical throughout the set of experiments."

line 20: "used IN the permeation source"

Agreed, we now replaced "as" by "in"

line 26: Eq. 5 ?

"Eq.5" should be "Eq.6", we now changed it.

line 26f: I understand background as "offset". But you are looking at instrumental noise?! The background, I would determine around each m/z = 0.5UMR, i.e. between two peaks. I also would try to determine the instrument noise there.

The "background" is misused, which should be the standard deviation of instrument noise. We replaced this by "$\sigma_{noise}$". Getting instrument noise from masses between two peaks is a good suggestion and we now applied both methods on the ambient dataset, and the difference is shown in Fig. 2 below. At masses above 600 Th, there is little difference between the two methods. At masses below 500 Th, in the "0.5UMR" method, the $\sigma_{noise}$ decreases, which might be caused by the counting algorithm when averaging the raw data, e.g. baseline removal.

line 31 and Fig. S3: I think, there is a trend of increasing "back ground" with decreasing m/z. 0.035 is background or instrumental noise, or detection limit? See my previous C4 comment.

There is indeed weak increase (~ 0.005 cps / 200 m/z) of $\sigma_{noise}$ (on both methods) with decreasing mass. We will rephrase the description of this figure. In this study, we took the median number (i.e. 0.035) over this mass range (800~1000). On the other hand, we consider this weak increase likely unimportant for the overall error estimate, because the analytical uncertainty ($\sigma_{ij}$) is overwhelming in most cases.

[Figure]

**Fig. 2**. Standard deviation (1 sigma) of signals using UMR data and "0.5UMR" data.

line 35: eq 6 ??

"Eq. 6" is now replaced by "Eq. 7"

line 51+ 54: "median over a short period of time (5 data points)"; does that mean over 25 min.? Then you might have indeed to consider influence by chemical changes!?

Yes, this corresponds to moving median over 25 minutes. We will add this information. Choosing the time window width is a tricky question, as the chemical changes do become an issue with too long averaging, but on the other hand taking a median (or average) of 3 points (15 mins) may conversely interpret measurement error (random noise) as chemical change.

Based on our experience, we would not expect chemical processes to cause such radical changes (in form of such peaks or dips that the moving median would miss) to major ion concentrations, in timescales of few minutes. Between the 5 and 7 points windows' there seems to be less variation, thus 5 points was considered to be a solid middle road for this purpose.

We have added a figure (Fig.S6) to the supplement, illustrating this effect and also hopefully clarifying the entire noise estimation method.

line 55f: I really don't understand what you did. Especially the last half sentence is unclear. Try to reformulate in clearer way.

We have reformulated this sentence.

line 64f: "dividing the "noise estimate" (i.e. signal minus trend) data into bins"; difficult to understand.

This is also reformulated and expanded on. A figure (Fig. S6) is added to illustrate the assignation procedure to signal bins.

line 69: S1-S9 (?), but you need 10 bins! From here on, you mix the notation "S1- S9" and the fact that your using 10 bins. Check text and figures for that and correct.

This was indeed a mistake in the text; although the highest 10% of signal was emitted, we still used 10 bins for the remaining data – this is now corrected.

line 73: 940? or 9084/10??

It should be the latter - also corrected this (typo).

---

## Author Comment (AC2) · 16 Aug 2016

P. Paatero (Referee)

Pentti.Paatero@helsinki.fi

This manuscript describes PMF analysis of a large matrix of time-of-flight spectra of ions formed of atmospheric VOC molecules. The main part of the ms is interpretation of the obtained six factors. This main part of the work is not discussed in this review.

The ms puts much emphasis in deriving reliable uncertainty estimates for the elements of the measured mass spectra. This is intended in order to provide firm foundation for PMF modeling of these measurements, also for analysis of future similar measurements.

The comparison of obtained uncertainty estimates with obtained residual values is badly erroneous. Hence, conclusions about quality of fit are also erroneous. I recommend that this manuscript should be published in ACP after this comparison is performed correctly and all text based on this comparison is rewritten according to the corrected comparison. Also, I request that all the numerous problems discussed below are corrected (or present text is enhanced so that correctness of present text becomes evident).

The abstract says

PMF was performed with a revised error estimation derived from laboratory data, and this approach was validated by mathematical diagnostics of the PMF solutions. Unfortunately, this statement is erroneous. Mathematical diagnostics indicate that the carefully derived uncertainty estimates of data values are in striking conflict with the residuals obtained by PMF modeling.

The main part of this review is concerned with this discrepancy. Additionally, here are remarks regarding different erroneous or questionable details in the presentation. Although I will present some criticism regarding the estimation of uncertainty estimates, I believe that these estimates are sufficiently accurate so that when the computed residuals are seen to be much larger than the estimated uncertainties, then the discrepancy is real, not caused by errors in uncertainty estimates.

Essential mathematical diagnostics are only found in the Supplement. No hint of them is found in the main text.

Section 3 of Supplement is "Examining Q distribution of time and variables" It is good that this data is presented. However, its interpretation was not right, as shown in the following.

IT IS ASSUMED THAT MODEL ASSUMPTIONS OF PMF DO HOLD

If model assumptions do hold, then residuals are only due to data noise, so that assumed data uncertainties agree with observed distributions of residuals. If model assumptions do hold,

then all profiles stay unchanged throughout the measurement period, and the assumed number of factors is right.

Notation: The dimensions of the matrix are m rows, n columns. There are p factors. Q sums over columns and rows are denoted by $Q_j$ and $Q_i$, respectively. When we say "residuals" we mean scaled residuals, i.e. residuals divided by respective assumed data uncertainties.

THEORETICAL VARIATION OF Qrow AND Qcolumn VALUES The Supplement says:

"The mean value of Q on all variables was well below 4, the threshold in robust-mode PMF. This suggests that all variables are well described by the model." These sentences confuses single point $Q_{ij}$ contributions with the overall Q sums ob- tained for an entire matrix, for an entire row, or for an entire column. It is possible (within model assumptions) that a few individual points get residuals >4, whereby the $Q_{ij}$ contributions from such points exceed 16.

Estimates for Q contributions due to columns or rows are obtained from Statistical theory. Approximately, theory says that the expected value of $Q_j$ from any column j is = m-p, and from any row i, $Q_i$ = n-p. It also says that approximately the statistical distribution of Q is equal to chi-squared distribution whose degrees of freedom is m-p for $Q_j$ and n-p for $Q_i$.

In the present work, p«n and p«m, thus we may approximate: Distributions are: chi2(n) for row Q's and chi2(m) for column Q's. As m and n are large, chi2 is well approximated by the normal distribution. Thus distributions of Q values are approximately:

for row Q's, distribution of $Q_i$ is N(n,sqrt(n))
for column Q's, distribution of $Q_j$ is N(m,sqrt(m))
where N(*,*) denotes normal distibution and sqrt(m) and sqrt(n) are the standard deviations of respective normal distributions. As an example, compute limits of these distributions for n=400, m=10000. Then the lower and upper 2-sigma limits for Q values, under model assumptions, are

360 to 440 for row Q's
9800 to 10200 for column Q's

It is seen that Q values come very close to their expected values m and n when model assumption are valid. Such "well-behaving" Q values are obtained in numerical simulations when the only simulated error is the random error in data values. If computed Q values deviate more in analyses of real data, then random noise in data values cannot be the explanation if assumed uncertainties are correct for data noise.

COMPARISONS WITH Q VALUES SHOWN IN SUPPLEMENT

There appears to be a conflict between parts a and b of Fig. S8. The overall averages of Q in parts a and b should be identical because they are averages of the same matrix of individual $Q_{ij}$ values. In the following discussion, we only consider part b of Fig. S8. which seems to agree with the value of Q/Qexp for 6 factors that is given for the overall Q in Fig 7 of the manuscript proper.

Distribution of Qrow values (fig S8/b) ranges from approximately 0.2n to 2n, some rows even exceed these limits. This variation is very much wider than the expected width of chi2 distribution for Qrow.

Thus it is concluded that model assumptions did not hold for this PMF modeling. The estimated noise in measured values does not explain the observed variation of Qrow from row to row. Small values of Qrow may have a simple explanation: censoring of values <DL (see below) has eliminated some variation from the data, so that Q contributions from BDL values are much less than expected. Thus Qrow values of low-intensity rows will be (much) smaller than expected.

Note that downweighting low-intensity ("weak" and "bad") columns will also decrease both Qrow and Qcolumn values below their expected values. On the other hand, there are significant numbers of rows with $Qrow > n + 4 \sqrt{n}$.

It is difficult to know the reasons for these large Qrow values. Possible reasons are e.g.:

- variation of component profiles from sample to sample.

- small slow variation of mass calibration and/or resolution from sample to sample.

- small variation of critical parameters of the ionization process, e.g. of temperatures

Careful study of residuals would be the first step in finding the reason(s) for increased Qrow and Qcol values.

It is essential to admit that "something" happens in the atmosphere, or in the measurement process, that is not currently understood.

One must not try to "explain away" this "something" by arguing that yes, this was already seen by others, there is nothing new in such variation of row and column Q values.

On the contrary, this is indeed not an exceptional case. It is a phenomenon that should be studied so that it is understood. If there is component profile variation, understanding this variation might be a significant step in understanding chemical processes in the atmosphere.

It is important to modify the manuscript so that this "something" is clearly presented. It is not necessary nor possible to determine in this manuscript the reasons behind the observed Q variation.

It might perhaps be good to discuss or mention possible reasons. Figure S8 is crucial in demonstrating this Q variation. It might be good to move Fig S8 to the main text. After all, determination of data uncertainties is one of the main contents of the paper. Importance of Fig S8 is based on carefully determined data uncertainties, thus it is not logical to hide Fig S8 in the Supplement.

It would be good to point out that the observed good overall Qexp values are misleading in this case. Some Qrow values are much too large while others are much too small, so that these two effects largely cancel each other in the value of overall Qexp. See also remark lines 306,307, below.

We would like to thank Pentti Paatero for giving the constructive, helpful and detailed comments and suggestions, especially for the discussion on the diagnostics of PMF solutions. These comments and suggestions greatly improved this manuscript.

According to the comments, we modified the manuscript in the following aspects:
1) Correct all problematic statements on the diagnostics of PMF solutions, and move the figure showing Q distribution to the main text, so that the "something" is clearly presented.
2) Improve the Supplementary Information by correcting typos and mistakes, and polish the language;

In the following, we respond to the referee's comments item by item.

DETAILED DISCUSSION OF THE MAIN PART

Eq(1) is unclear. What quantities are represented by [X] and by the numerator and denominator. The text says

"the numerator on the right hand side is the sum of all detected ions" This probably means: ... is the sum of detected ion concentrations?

Further: "the denominator is the sum of all reagent ion signals" What does this mean? Note that there are no square brackets in the denominator.

We simplified this equation to

$$[HOM] = \frac{HOM(NO_3^-)}{\sum_{i=0}^{2}(HNO_3)_i(NO_3^-)} \times C$$

For each HOM molecule, its concentration ([HOM]) equals to its respective detected signal normalized by the summed reagent ion signal, multiplied by the calibration coefficient $C$. We also clarified the related text.

On lines 126, 132, 176, and possibly elsewhere, PMF is called "an algorithm". This is wrong. PMF is a model, it defines the equations that should be fulfilled by the computed factor elements. Algorithm is a procedure for finding the values for factor elements so that they fulfill the model. There are currently at least 4 different algorithms for fitting or "solving" this model PMF. Please use correct terminology! Admittedly, the majority of chemically oriented papers do not pay attention to this distinction. In fact, it would be good to specify which PMF algorithm was used: the original algorithm in PMF2, or a PMF script executed by program ME-2? There are slight differences between these programs, especially if there is rotational ambiguity in the model. (There is probably very little rotational ambiguity in this work, so that the distinction PMF2 vs. ME-2 does not matter now. However, it is good manners to specify the used tools.)

Agreed. We corrected the terminology. Sofi5.2 is based on program ME-2, we specified this in line 132-134.

Lines 167 - 169 say "... Therefore, the data matrix used in this work is in unit-mass resolution, and peak fitting was performed afterwards to identify the elemental formula of peaks..."

This is an important decision and probably quite suitable for these data sets. There would also be other ways of formulating the matrix. It might be useful to learn whether the authors experimented with different ways, and what considerations lead them to select the unit-mass resolution. However, if the authors plan to examine this question later in more detail in another paper, then it is ok to not discuss this question now.

While the use of UMR data here does not make full use of the acquired HR spectra, the determination of errors for HR fits becomes much more complex. Especially in this case, when our signals are low, we would first need to average to at least 1-hour data in order to get most peaks smooth enough for reliable fitting. In addition, even assuming all peaks are smooth enough, the presicion of peak fitting on overlapping peaks shows complicated dependence on mass calibration, resolving power, and peak intensity, as studied by Cubison and Jimenez (2015). Thus we chose UMR data as this already gave us much new insights into the HOM formation pathways.

Cubison, M. J. and Jimenez, J. L.: Statistical precision of the intensities retrieved from constrained fitting of overlapping peaks in high-resolution mass spectra, Atmos. Meas. Tech., (8)2333-2345, 2015.

Lines 184-186 say " I is the signal strength (ions/second) of the ion, ts is the integration time in seconds, and a is a factor accounting for the fact that a single ion will generate a Gaussian-shaped pulse in the detector, rather than a single peak. " Here is confusion (or sloppy wording). I am not sure how to understand this topic.

First, I believe that the words "generate a Gaussian-shaped pulse in the detector" are a mistake. The intention is probably to say that "individual ions produce pulses whose pulse height distribution is of Gaussian shape."

Second, why does the pulse height distribution matter at all? If ions are not actually counted but count rate is determined by integrating the current that is due to accumulated pulses, then the statement would be understandable: the variation of charge produced by each ion does indeed contribute to the uncertainty of integrated current. In contrast, if ion pulses are actually counted (I believe this is the case), then the variation of pulse height from ion to ion does not directly contribute to uncertainty, except if the variation is large enough so that a fraction of ions are not counted at all. Please clarify or correct.

This sentence meant to say that the signal generated by a single ion in the micro-channel plate (MCP) detector is not constant but follows Gaussian distribution, which also contributes to the overall imprecision. The referee is right, since the CI-APi-TOF used in this work was using a time-to-digital converter as the data acquisition card, the Gaussian distribution of signal height should not affect the accounting statistics, as all signals were amplified to be large enough to cross the threshold. The factor $a$ was empirically determined by fitting the lab data, and it might arise from some unaccounted uncertainty in the data, such as shifting of mass calibration, baseline correction. We have corrected the statement in the main text. Keeping the empirical factor $a$ also makes the equation (Eq.7) applicable regardless of the type of the data acquisition card.

Section 2.3.2. It would be good to state clearly that the data matrix consists of counts-per-second values, obtained as 5-minute averages. This fact can be inferred from the present text but why not help the reader by stating it explicitly.

Agreed, we now stated it explicitly. We slightly modified the sentence to "These input data matrix consists of counts-per-second (cps) values, averaged from raw data using 5-minute time resolution, and a total number of 9084 mass spectra were then obtained".

The paragraph beginning on line 209 claims that Paatero et al (2003) recommended censoring variables that are below DL by fixing the values at DL/3 and uncertainty at DL. This claim is entirely fictitious and wrong. There is not a single word suggesting censoring BDL values in the 2003 paper.

In contrast, certain PMF-related papers advise against this practice. There is no demonstrated benefit from this practice, provided that low S/N (i.e. "weak" and "bad") variables (entire columns) are downweighted as recommended in that 2003 paper. On the other hand, my personal experience in reviewing has revealed several cases where such censoring created one or two ghost factors, i.e. numerical artefacts caused by censoring. Also, censoring often creates bias in the results.

It is possible (likely?) that in the present case, censoring BDL values did not cause noticeable harm in main results because there were so many strong variables. The only likely harm might be that some details were lost from those columns where itensies are lowest. Thus it is not reasonable to suggest that the work should be redone using the original measured BDL values and uncertainties. On the other hand, the present formulation of the paper would be interpreted by your readers as a rule saying that BDL values -must- be censored. In order to help prudent practices prevail among atmospheric scientists, I request the following addition in the ms:

After explaining that BDL values were replaced by DL/3 in this work, you shall insert a remark, something like the following:

After this work was completed, we became aware that this practice of replacing BDL values by fixed values is harmful and provides no advantage at all, although in this specific case, the main results were possibly not harmed. Thus we emphasize that in future studies, our example should not be followed. Instead, values < DL and their uncertainties should remain unchanged in data and error matrices.

This was a wrong citation. The approach we used in this work is similar to what has been suggested by Polissar et al. (1998). We corrected the citation. We also realized that there was another mistake that we actually replaced corresponding error with 2DL ($6\sigma_{noise}$) instead of DL ($3\sigma_{noise}$). This is to make those data as "bad signals". For the variables that contains many censored data points, very likely they will be further downweighted by 10-fold. We also corrected this in the manuscript and the supplement.

We checked the effect of censoring data in our case by running PMF with uncensored data. We evaluated the similarity of the results by checking the correlation between solutions using censored data and those using uncensored data. The results are provided in Table S1. From 2

factors to 6 factors, PMF solutions are almost identical (UC>0.99). Similar to what is expected, censoring data did not affect the results in our case. Even so, we do agree that censoring data is probably not a good practice. We will add some discussion in supplementary information S3, where we recommend readers not censoring data in future works.

Signal-to-noise estimates (S/N) are discussed in the paragraph beginning on line 209. Please state which formulation of S/N was used. There are two published formulations:

(1) the recommended S/N definition, distributed with EPA PMF v5 and published in the Supplement of

Brown, S. G., Eberly, S., Paatero, P. & Norris, G. A., Methods for estimating uncertainty in PMF solutions: Examples with ambient air and water quality data and guidance on reporting PMF results. Science of the Total Environment. 518-519, p. 626-635, 2015

(2) the earlier problematic S/N definition, suggested in the quoted 2003 paper and used in all earlier EPA PMF versions.

The numerical values produced by these two methods differ from each other, thus the readers need to know which method was used by you: one of these, or your own method (define).

The Sofi5.2 program uses the earlier S/N definition suggested in Paatero et al. (2003). We now mention this in the main text.

The manuscript mentions that some columns were downweighted (DW) by 2 or by 10. State how many columns were DW by 2 and by 10. Numbers of DW columns also influence the Qexp values. When you reported Q/Qexp, did you use correct Qexp values that take this influence into account? If not, state this clearly! If yes, state that, too!

The influence of DW columns on $Q_{exp}$ value has been taken into account in $Q/Q_{exp}$ values. We have downweighted 173 variables as weak signals and 152 variables as bad signals. Fig. S9 in the revised supplement shows the distribution of signal-to-noise ratio, and we added this information in the main text.

Lines 306, 307 say "From two to seven factors, Q/Qexp decreases stepwise from 2.44 to 0.76. The closeness to unity indicates that the estimated error is appropriate for the model."

This good agreement of Qexp with its theoretical value is misleading (see above, too). Because of downweighting and/or censoring, low concentrations contributed to Qexp much less than expected. This is OK once it is recognized. However, there are also high concentration values that contribute to Qexp much more than expected, so that the overall Qexp appears acceptable. Thus it is not right to claim that the estimated error is appropriate for this PMF modeling with 6 or 7 factors.

Agreed. We re-wrote this section and corrected all wrong statements concerning $Q/Q_{exp}$.

DETAILED DISCUSSION OF THE SUPPLEMENT

There are too many typos and broken sentences, poor language, and even mistakes in the equations. It must be emphasized that all text, even the Supplement, should be carefully checked by one or two of senior authors before the manuscript is submitted. Language must be improved in the whole of Supplement text. This is essential in order that the text may be understood!

lines 9, 10, 35

These lines refer to "a" in Eq(6). There is no "a" in Eq(6)? Confusion? Is Eq(7) intended? This confusion may be present elsewhere, too

The referee is correct, Eq. 6 should be Eq.7. Other similar mistakes were also corrected, and the language of the Supplement was considerably improved.

line 33 says

"also confirms the validity of the pre-assumption" which pre-assumption? Do not pose extra difficulties for the reader, please be explicit!

This sentence was indeed confusing. We removed this sentence and rephrased the whole paragraph.

line 64 says

"by dividing the "noise estimate" (i.e. signal minus trend) data" what does "trend" mean here. I cannot even guess.

We now changed the sentence to "…by dividing the "noise estimate" (i.e. signal minus moving median) data into bins according to their signal (cps)" to be more clear.

lines 64 to 70

This paragraph is almost impossible to understand. Equations are the language of mathematics. Please use equations as the main method of defining what was done. Verbal explanations may only be used as a help for understanding the equations.

The paragraph has been revised and a figure added to illustrate the signal binning procedure.

lines 82 to 89

This paragraph confuses superposition and convolution. First do the math properly, then rewrite the paragraph.

Quite right. As this paragraph had redundant verbal explanation of Eq. 6 to Eq. 8 and Eq. S1, and only otherwise stated the obvious, we have omitted it for clarity.

Fig S1, caption contains: "All the flows were set identical throughout the experiments" better to write: "All the flows were kept unchanged throughout the experiments"
or "All the flows were constant throughout the experiments"

Agreed, we changed it to "All the flows were constant throughout the experiments"

Eq(S1)

This equation contains a parameter "a". The "Allan equation" Eq(7) in the main text also contains a parameter "a". However, the equations are different, and the "a" values are thus different, too. In Eq(7), the "a" is dimensionless and approximately =1. In Eq(S1), the "a" has dimension and its value depends on integration time.

This confuses the reader significantly and quite unnecessarily. Please rewrite the supplement so that the same equation is used in both texts.

We agree, this was confusing. We now re-labeled $a$ in the supplement equations and text to $c$, and added a sentence to explain its relation to the Allan equation.

lines 103-104 say

"Parameter "a" is similar the "a" in the Allan et al. (2003) equation".

This statement is badly misleading, as already noted, above. If Eq (S1) is not changed as I suggested, then this statement must be changed to its opposite, warning the reader that the two symbols "a" are not the same.

Corrected, same as above.

Eq(S2): There is a problem with this equation. I suspect that an equals sign is missing before the square root.
The referee is correct. This was amended.

Eq(S3)

I do not understand this equation at all. What are the X symbols? What is the equation trying to say?

This equation was supposed to contain the final numerical values for c, e and their errors. It is now corrected in Eq.S2.

Fig S7.

These confidence limits for sigma values do not make sense. There must be some problem in their evaluation. Possibly, an invalid estimation principle was used.

We checked the error values and there was indeed a problem - with plotting, absolute conf int. (min/max) values were plotted as error bar height (from centre), making them way too large. Now corrected.

Fig S8

I assume that s8/a represents the 6-factor solution (why is it not stated?). Then Figs a and b are based on same Q_ij values. However S8/b and S8/a of Q/Qexp seem to be in conflict: there are many more values >1 in S8/b than in S8/a. Is there a natural explanation (show the explanation if there is one) or is there an error in generating the figures?

Both sub figures are intended for evaluating 6-factor solution, we now explicitly mention this in the figure caption. The original Fig.S8a suffers a mistake in generating the figure. Also in original Fig.S8a&b, we used the estimated error before downweighting to calculate the $Q_{ij}$, and this causes a discrepance in comparison to the overall $Q/Q_{exp}$ given by Sofi using the error after downweighting. We corrected both mistakes.  As suggested in the general comments, we now moved the Q distribution figure to the main text as Fig. 7b&c.

Fig S9
The caption says "... the purple one is the residue ..." The correct term in numerical context is "residual". In chemistry, "residue" might be used for remains of substances.

This is now corrected.